# TabPFN Unleashed: A Scalable and Effective Solution to Tabular Classification Problems

**Si-Yang Liu** [1 2]   **Han-Jia Ye** [1 2]

## Abstract

TabPFN has emerged as a promising in-context learning model for tabular data, capable of directly predicting the labels of test samples given labeled training examples. It has demonstrated competitive performance, particularly on small-scale classification tasks. However, despite its effectiveness, TabPFN still requires further refinement in several areas, including aligning with downstream datasets, handling high-dimensional features, and scaling to larger datasets. In this paper, we revisit existing variants of TabPFN and observe that most approaches focus either on reducing bias or variance, often neglecting the need to address the other side, while also increasing inference overhead. To fill this gap, we propose BETA (**B**agging and **E**ncoder-based Fine-tuning for **T**abPFN **A**daptation), a novel and effective method designed to *minimize both bias and variance*. To reduce bias, we introduce a lightweight encoder to better align downstream tasks with the pre-trained TabPFN. By increasing the number of encoders in a lightweight manner, BETA mitigate variance, thereby further improving the performance of the pretrained TabPFN. Additionally, bootstrapped sampling is employed to further reduce the impact of data perturbations on the model, all while maintaining computational efficiency during inference. Our approach enhances TabPFN's ability to handle high-dimensional data and scale to larger datasets. Experimental results on over 200 benchmark classification datasets demonstrate that BETA either outperforms or matches state-of-the-art methods.

## 1. Introduction

Tabular data is one of the most widely used data formats across various domains, including finance (Cao & Tay, 2001), healthcare (Hassan et al., 2020), e-commerce (Nederstigt et al., 2014), and medical analysis (Schwartz et al., 2007; Subasi, 2012). Despite its ubiquity, modeling tabular data with deep learning methods remains a challenge due to its heterogeneous nature (Borisov et al., 2024b). Yet recent advancements have led to the development of tabular foundation models (van Breugel & van der Schaar, 2024) , such as TabPFN (Tabular Prior-Fitted Networks) (Hollmann et al., 2023; 2025). TabPFN operates in two stages: *pre-training* and *inference*. During the pre-training stage, the model is pre-trained on a diverse set of synthetic datasets. In the inference stage, given a new task and a set of labeled examples as a "prompt," TabPFN directly predicts the labels of test samples using in-context learning, without requiring further parameter updates. This approach enables TabPFN to achieve performance comparable to or even surpass tree-based methods, particularly on small tabular datasets (McElfresh et al., 2023).

TabPFN has shown potential across a wide range of applications, including tabular data generation (Ma et al., 2024a; Margeloiu et al., 2024b), data augmentation (Margeloiu et al., 2024a), and time series forecasting (Hoo et al., 2025). These use cases highlight the versatility of TabPFN, positioning it as a model worthy of further exploration. However, alongside application-driven studies, there is a growing interest in improving TabPFN's performance from various perspectives (Feuer et al., 2024; Thomas et al., 2024; Xu et al., 2025; Ma et al., 2024b). While previous research has reported performance improvements, the underlying reasons driving these gains remain unclear. These advancements are often fragmented, with each approach focusing on a specific aspect, and some methods even sacrifice efficiency for enhanced performance, without a comprehensive understanding of how to improve TabPFN systematically.

In this paper, we adopt the bias-variance decomposition framework introduced by Nagler (2023) to analyze the generalization error of TabPFN and its variants. This framework allows us to revisit and categorize existing methods, revealing that performance improvements typically arise

[1] School of Artificial Intelligence, Nanjing University, China
[2] National Key Laboratory for Novel Software Technology, Nanjing University, China. Correspondence to: Han-Jia Ye <yehj@lamda.nju.edu.cn>.

*Proceedings of the 42nd International Conference on Machine Learning*, Vancouver, Canada. PMLR 267, 2025. Copyright 2025 by the author(s).

from addressing either bias or variance. However, these approaches often neglect the other aspect, leading to suboptimal performance. Therefore, there is a need for a method that simultaneously addresses both bias and variance to improve performance.

To this end, we propose a novel, effective, and scalable approach, BETA. Our method enhances TabPFN's performance by introducing a *fine-tuning* stage, enabling parameter-efficient adaptation that aligns the downstream dataset distribution with the pre-trained TabPFN to mitigate bias. To further reduce variance, BETA maps raw data into multiple latent spaces and employs computationally efficient bootstrapped sampling during inference. Beyond performance improvements, BETA offers a lightweight and scalable solution that effectively handles high-dimensional data and large datasets while maintaining inference efficiency.

BETA improves performance by refining both the fine-tuning and inference stages to reduce bias and variance. In the fine-tuning phase, we employ a lightweight encoder module as an input feature adapter, which transforms datasets with arbitrary dimensionality into multiple fixed-dimensional representations, thereby naturally enabling dimensionality reduction. To further enhance generalization, we integrate Batch Ensemble (Wen et al., 2020; Gorishniy et al., 2025) to increase the number of encoders in a lightweight manner, which introduces diversity in learned representations and reduces variance. These enhancements enable BETA to improve robustness and scalability, ensuring better adaptation to downstream datasets. In the inference phase, we introduce bootstrapped sampling, a technique that has been largely overlooked in previous TabPFN variants. By generating multiple subsets of the dataset as support sets for the context composition, BETA further reduces variance. Furthermore, our method seamlessly integrates with Error-Correcting Output Codes (Dietterich & Bakiri, 1995) (ECOC) to effectively handle multiclass classification tasks with more than 10 classes.

Experimental results on multiple benchmark datasets, including over 200 classification tasks, demonstrate that our method significantly improves the performance of TabPFN. We describe our main contributions below.

1. We introduce an adaptation method for TabPFN that addresses key limitations related to dataset size, high-dimensional features, and multiclass classification tasks.

2. By analyzing the generalization error of existing TabPFN variants through bias-variance decomposition and experiments on real-world datasets, we introduce BETA, a method that effectively mitigates both bias and variance.

3. We achieve state-of-the-art performance on the largest benchmark datasets to date, demonstrating the robustness and scalability of our method for real-world tabular tasks.

The code is available at https://github.com/LAMDA-Tabular/BETA.

*Remark*. We have noticed that the latest release of TabPFN-v2 (Hollmann et al., 2025) has partially alleviated some of the limitations previously mentioned. Specifically, TabPFN-v2 incorporates design improvements that enable it to handle larger datasets and more features. However, it is important to highlight that TabPFN-v2 is a concurrent work, and while it partially mitigates these limitations, it does not fully resolve the challenges posed by dataset size and feature count. Thus, many of the improvements proposed in this paper are general enhancements that can complement TabPFN-v2 and potentially further its applicability to a broader range of tasks.

## 2. Preliminary

In this section, we provide a brief overview of TabPFN and analyze its properties, while also revisiting existing variants.

### 2.1. TabPFN

We consider a tabular dataset consisting of $N$ examples and $d$ features. Each instance $\boldsymbol{x}_i \in \mathbb{R}^d$ is represented by $d$ feature values, where $\boldsymbol{x}_{i,j}$ denotes the $j$-th feature of instance $\boldsymbol{x}_i$. These features can be numerical ($\boldsymbol{x}_{i,j}^{\text{num}} \in \mathbb{R}$) or categorical ($\boldsymbol{x}_{i,j}^{\text{cat}}$), with categorical values often encoded as integers. Each instance is associated with a label $\boldsymbol{y}_i$, where $\boldsymbol{y}_i \in [C] = \{1, \ldots, C\}$ for classification task, and $\boldsymbol{y}_i \in \mathbb{R}$ for regression task. Given a training dataset, $D_{\text{train}} = \{(\boldsymbol{x}_{\text{train}}^{(i)}, \boldsymbol{y}_{\text{train}}^{(i)})\}_{i=1}^{N_{\text{train}}}$, and test samples, $\boldsymbol{X}_{\text{test}} = [\boldsymbol{x}_{\text{test}}^{(i)}]_{i=1}^{N_{\text{test}}}$, the goal is to predict the corresponding labels, $\boldsymbol{Y}_{\text{test}} = [\boldsymbol{y}_{\text{test}}^{(i)}]_{i=1}^{N_{\text{test}}}$, as accurately as possible.

TabPFN follows a two-stage process: a pre-training stage, where the model is trained on synthetic datasets by minimizing the discrepancy between the predicted label of the test instance and its true label, and an inference stage, where it directly predicts the labels of test samples given a set of labeled training examples.

Following pre-training, TabPFN takes the entire training dataset $D_{\text{train}} = (\boldsymbol{X}_{\text{train}}, \boldsymbol{y}_{\text{train}})$ and the features of query points $\boldsymbol{X}_{\text{q}}$ as context to make predictions. This is done using **PFN-Style Batching** (Hollmann et al., 2023), where $N_{\text{test}}$ test samples are batched into a single "prompt," as illustrated in Figure 1. Since TabPFN is trained with a fixed input dimensionality, typically set to a predefined value $d_{\max}$ (*e.g.*, 100), datasets with fewer features ($d < d_{\max}$) need to be extended via zero-padding before being processed. Formally, given an input instance $\boldsymbol{x}_i \in \mathbb{R}^d$, the padded input $\tilde{\boldsymbol{x}}_i \in \mathbb{R}^{d_{\max}}$ is obtained as:

$$\tilde{\boldsymbol{x}}_i = [\boldsymbol{x}_i, \boldsymbol{0}] \in \mathbb{R}^{d_{\max}},$$

where $\boldsymbol{0} \in \mathbb{R}^{d_{\max}-d}$ represents the zero-padding applied to match the required input dimensionality. However, if a dataset exceeds this limit ($d > d_{\max}$), TabPFN cannot directly process such datasets, as its architecture is not de-

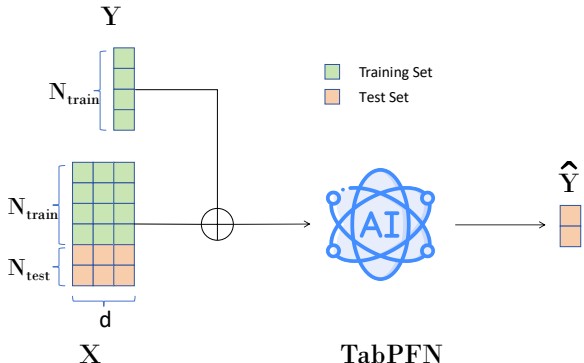

Figure 1: The inference process of TabPFN. The training set and test set are concatenated, with the training labels added to the input sequence. TabPFN then performs a single forward pass to generate predictions for all test samples.

signed to accommodate higher-dimensional inputs.

TabPFN outputs a probability distribution over possible labels $y_q \in \{1, \ldots, C\}$. Specifically, let $q_\theta$ denote the logits produced by TabPFN, where $\theta$ represents the parameters of the pre-trained model. The posterior predictive distribution is given by:

$$p_\theta(y_q \mid X_q, D_{\text{train}}) = \frac{\exp(q_\theta(\tilde{X}_q, \tilde{D}_{\text{train}})_{[y_q]})}{\sum_{c=1}^{C} \exp(q_\theta(\tilde{X}_q, \tilde{D}_{\text{train}})_{[c]})}, \tag{1}$$

where $\tilde{D}_{\text{train}}$ denotes the zero-padded training dataset, acting as the support set for the context composition. Since TabPFN performs inference directly, the model typically applies multiple rounds of feature shuffling on the original features, followed by prediction. The final prediction is obtained by averaging the results from these multiple inferences. For more details about how the model processes inputs, please refer to Appendix B.4.

## 2.2. TabPFN Variants

Existing research to improve the performance of TabPFN has primarily focused on two different strategies, stemming from its dependency on $q_\theta$ and $D_{\text{train}}$ in Equation 1. One approach optimizes context selection, refining $D_{\text{train}}$ to provide more informative support examples. The other concentrates on fine-tuning TabPFN, improving $q_\theta$ to better adapt to downstream tasks. These two strategies offer complementary solutions for enhancing TabPFN's overall performance. **Context Selection for Scaling TabPFN.** The transformer architecture in TabPFN inherently leads to quadratic growth in memory usage as the context length increases. As a result, the size of the support set used as context in each prediction is limited. Meanwhile, a well-chosen $D_{\text{train}}$ provides more relevant support examples, improving generalization. A simple approach is random subsampling (McElfresh et al., 2023), but this can degrade performance, especially on large datasets (Ma et al., 2024c). More structured methods include

sketching techniques, such as CoreSet, K-Means, and Data Distillation (Ma et al., 2024c), which aim to compress large datasets into representative subsets while preserving essential information (Feuer et al., 2023). TuneTables (Feuer et al., 2024), on the other hand, attempts to encode the dataset into a compact learned representation, reducing memory overhead. Another class of methods selects sample-specific contexts instead of a fixed support set. For instance, MixturePFN (Xu et al., 2025) partitions the training set into multiple subsets and assigns the most relevant one to each test sample. Similarity, LocalPFN (Thomas et al., 2024) and TabDPT (Ma et al., 2024b), instead use nearest neighbors to dynamically construct the support set $D_{\text{train}}$ for each query (KNN-based selection). While these approaches improve TabPFN's performance by refining $D_{\text{train}}$, they disrupt PFN-style batching, significantly reducing inference efficiency.

**Fine-tuning Strategies for Enhancing TabPFN Performance.** The second approach to improving TabPFN's performance focuses on fine-tuning the pre-trained model to better adapt to downstream datasets. As shown in Equation 1, the predictive distribution $p_\theta(y_q \mid X_q, D_{\text{train}})$ depends on both the training set $D_{\text{train}}$ and the model parameters $\theta$. While optimizing $D_{\text{train}}$ improves the quality of support examples, fine-tuning enhances $q_\theta$, enabling better alignment with downstream tasks. A straightforward approach is to fine-tune all model parameters, as seen in TabForestPFN (den Breejen et al., 2024) and LocalPFN (Thomas et al., 2024), which improves performance by adapting TabPFN's learned prior to specific datasets. However, full fine-tuning incurs substantial computational costs due to the large number of model parameters. To mitigate this, TuneTables (Feuer et al., 2024) offers a more efficient alternative by applying prompt-tuning (Lester et al., 2021) for classification tasks, which adjusts only a small set of parameters, and fine-tuning the entire model for regression tasks. Another efficient approach is adapter-based fine-tuning, as employed in MixturePFN (Xu et al., 2025), which fine-tunes additional adapter layers (Houlsby et al., 2019) rather than modifying the entire model. A detailed related work is presented in Appendix A.

## 2.3. Generalization Analysis of TabPFN Variants

While various TabPFN variants have been proposed to enhance performance, their underlying mechanisms remain insufficiently understood. In this section, we leverage the **bias-variance decomposition framework**, as described in Equation 4, which is introduced by Nagler (2023), to analyze the generalization error of TabPFN and its variants on two real-world datasets: *Adult* (Barry & Ronny, 1996) and *Bank* (S. et al., 2014). To assess the impact of different strategies, and based on our previous analysis, we evaluate the original TabPFN (without context length restrictions) alongside sev-

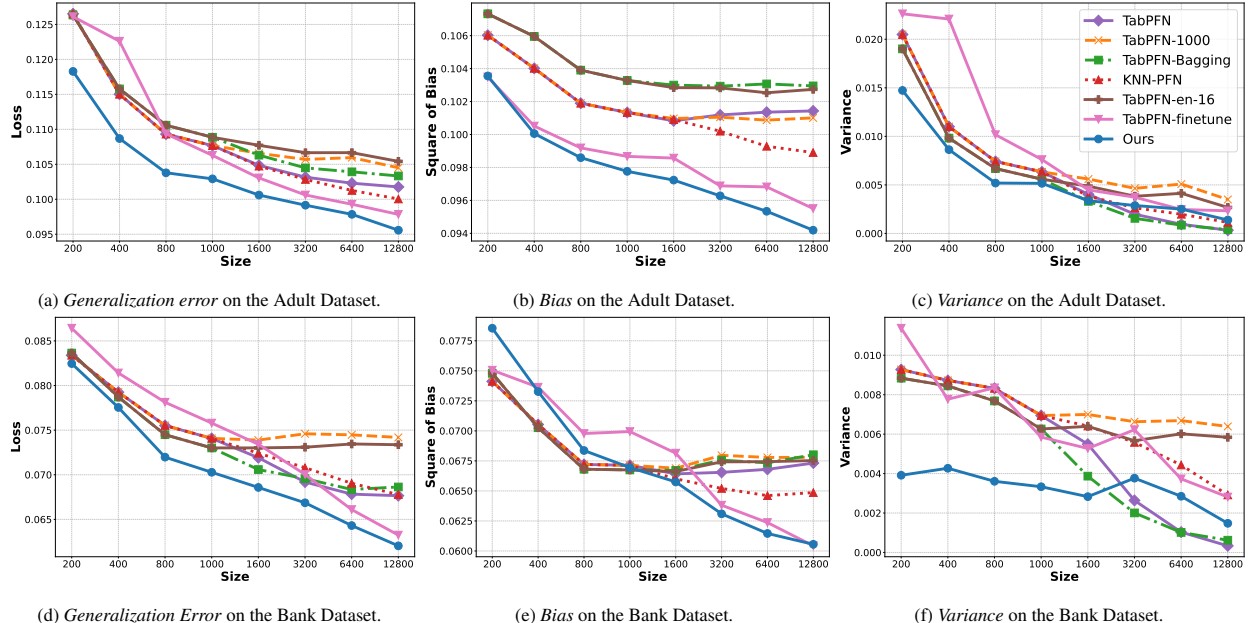

(a) *Generalization error* on the Adult Dataset.      (b) *Bias* on the Adult Dataset.      (c) *Variance* on the Adult Dataset.

(d) *Generalization Error* on the Bank Dataset.      (e) *Bias* on the Bank Dataset.      (f) *Variance* on the Bank Dataset.

Figure 2: Generalization Error, Bias, and Variance for different methods on the *Adult* and *Bank* datasets. The methods shown include TabPFN-1000 (subsample size = 1000), TabPFN, TabPFN-en-16, TabPFN-Bagging, KNN-PFN, TabPFN-finetune, and Ours (BETA). The legend is located in the top-right plot for clarity.

eral representative variants. These include TabPFN-1000 (random subsampling of 1000 samples), TabPFN-en-16 (randomly sample once and perform feature shuffling 16 times to form an ensemble of 16 predictions), KNN-PFN (KNN-based context selection), TabPFN-finetune (full model fine-tuning), and TabPFN-Bagging (bootstrapped sampling with 16 varying contexts). TabPFN-Bagging, unlike ensemble methods that require training multiple independent models, employs bootstrapped sampling to construct diverse support sets within a single inference process. A detailed description of its implementation is provided in Appendix B.5.

To further contextualize these findings, we compare all evaluated methods against our proposed approach, BETA. The experimental results, presented in Figure 2, reveal several key trends in the bias-variance tradeoff across different TabPFN variants. To complement the visual analysis, Table 3 reports the relative percentage changes in bias and variance (negative values indicate reduction) for several representative variants, averaged across dataset sizes from Figure 2, for both the *Adult* and *Bank* datasets. It can be observed that, except for our method, none of the representative TabPFN variants manage to simultaneously reduce both bias and variance across the two datasets.

As shown in Figure 2 (b,c,e,f), increasing context length reduces variance while bias plateaus, consistent with prior findings (Nagler, 2023). Similarly, KNN-based context selection reduces bias but increases variance compared to using the full dataset, as it relies on localized subsets. Since these trends align with previous work, we focus on addi-

tional findings.

**1) Fine-Tuning and its Impact:** Figure 2 (b,e) shows that fine-tuning (TabPFN-finetune) effectively reduces bias by aligning the model's prior with the characteristics of the dataset. However, it also increases variance, especially when the training set is small, where overfitting amplifies prediction variability, as shown in Figure 2 (c,f). These results suggest that fine-tuning requires careful regularization to balance bias and variance.

**2) Ensemble Strategy and Bias-Variance Tradeoff:** Figure 2 (b,c) indicates that the ensemble-based method (TabPFN-en-16) reduces variance but may increase bias due to unsupervised feature transformations. This observation highlights that ensemble-style TabPFN, while helpful in reducing variance, may require additional bias-reducing strategies to optimize overall model performance.

**3) Effectiveness of Bagging:** As shown in Figure 2 (c,f), Bagging significantly reduces variance while maintaining stable bias. By introducing diversity through bootstrapped sampling, it achieves variance reduction comparable to ensemble methods but at a lower computational cost. These results highlight Bagging as a simple yet effective approach for improving the performance of TabPFN.

The results above, based on additional findings not covered in (Nagler, 2023), indicate that TabPFN variants typically impact either bias or variance, but rarely both simultaneously. For instance, KNN-based context selection reduces bias but increases variance, while the original ensemble

Table 1: Comparison of TabPFN and related methods in terms of their impact on bias and variance, as well as their effectiveness in handling large datasets, high-dimensional features, adaptability to multiclass classification, and inference efficiency. The compared methods include TabPFN (Hollmann et al., 2023), TuneTables (Feuer et al., 2024), TabForestPFN (den Breejen et al., 2024), LocalPFN (Thomas et al., 2024), and MixturePFN (Xu et al., 2025). Our proposed method (BETA) is distinguished for its ability to balance bias and variance while maintaining efficient scaling, adaptability to multiclass classification, and lightweight fine-tuning. Different row colors indicate the *strategies used for improving performance* ( ), *scalability* ( ), and *efficiency* ( ). The "No Additional Inference Cost" entrie refers to computational efficiency relative to the ensemble-style TabPFN setting commonly used in practice, as discussed in subsection 2.1.

| | BETA (Ours) | TabPFN | TuneTables | TabForestPFN | LocaLPFN | MixturePFN |
|---|---|---|---|---|---|---|
| Scales to Large Datasets | ✓ | ✗ | ✓ | ✗ | ✓ | ✓ |
| Handles High-Dimensional Data | ✓ | ✗ | ✗ | ✗ | ✗ | ✗ |
| Adapts to More Than 10 Classes | ✓ | ✗ | ✓ | ✗ | ✗ | ✗ |
| Reduces Bias | ✓ | ✗ | ✓ | ✓ | ✓ | ✓ |
| Reduces Variance | ✓ | ✗ | ✗ | ✗ | ✗ | ✗ |
| No Additional Inference Cost | ✓ | ✓ | ✓ | ✓ | ✗ | ✗ |
| Fine-Tuning | lightweight encoder | ✗ | prompt & backbone | backbone | backbone | adapter |

strategy in Hollmann et al. (2023) (TabPFN-en-16) lowers variance but may increase bias. These observations highlight the need for a method that jointly optimizes both aspects, motivating the development of BETA.

Existing approaches to improving TabPFN focus on context selection or fine-tuning, but often introduce trade-offs in computational efficiency, scalability, and adaptability. A comparison in Table 1 highlights how different strategies affect bias, variance, and their ability to handle large datasets and high-dimensional features efficiently.

## 3. BETA

To address both bias and variance issues of TabPFN observed during the previous experiments, we propose a unified strategy for improving the performance of TabPFN. In addition to the inference stage, we introduce a fine-tuning stage. Our approach improves performance in both fine-tuning and inference, as shown in Figure 3. During fine-tuning, we refine input representations to better align the downstream data distribution with the pre-trained TabPFN, reducing both bias and variance. In the inference stage, we incorporate bootstrapped sampling to further reduce variance without additional computational overhead, compared to the ensemble-style TabPFN setting, as discussed in subsection 2.1.

**Minimizing Bias with Encoder-Based *Fine-Tuning*.** Our analysis in subsection 2.3 highlights the importance of minimizing bias for improved generalization. To achieve this, we introduce a lightweight encoder while keeping the pre-trained TabPFN parameters frozen. This encoder transforms raw input features into a latent space that better aligns the pretrained TabPFN with the downstream data distribution.

Let $\boldsymbol{x}_i \in \mathbb{R}^d$ denote the raw feature vector, and let $\boldsymbol{E}_\Phi$

represent the encoder with parameters $\Phi$. To enhance expressiveness and incorporate nonlinearity, these layers are constructed using a sequence of operations as described by Gorishniy et al. (2021):

$$\boldsymbol{E}_\Phi(\boldsymbol{x}) = \text{Linear}\left(\text{Dropout}\left(\text{ReLU}\left(\text{Linear}\left(x\right)\right)\right)\right). \quad (2)$$

This transformation can be extended by stacking multiple such blocks, allowing the encoder to learn hierarchical representations suited for complex downstream tasks.

Given a query set $\boldsymbol{X}_q$ and a support set $(\boldsymbol{X}_s, \boldsymbol{y}_s)$, we define their respective latent representations as:

$$\boldsymbol{Z}_q = \boldsymbol{E}_\Phi(\boldsymbol{X}_q), \quad \boldsymbol{Z}_s = \boldsymbol{E}_\Phi(\boldsymbol{X}_s).$$

The posterior predictive distribution over the query labels is then expressed as:

$$q_\theta\left(\boldsymbol{y}_q \mid \boldsymbol{Z}_q, (\boldsymbol{Z}_s, \boldsymbol{y}_s)\right).$$

This formulation ensures that the encoder effectively maps raw features into a structured space, enabling TabPFN to better capture relevant patterns while mitigating bias.

This fine-tuning method provides a lightweight and effective approach to reduce bias by better aligning the model's prior with the downstream tasks. It enables end-to-end fine-tuning, facilitating a more efficient adaptation of the pre-trained TabPFN to the downstream tasks.

**Mitigating Variance with Multiple Encoders.** To further mitigate variance, we introduce multiple encoders, each learning a distinct transformation of the input data. By jointly training these encoders while keeping the pre-trained TabPFN parameters $\theta$ frozen, the model captures diverse feature representations, reducing variance.

For the $k$-th encoder, the support and query set representations are encoded as follows:

$$\boldsymbol{Z}_s^{(k)} = \left[\boldsymbol{E}_\Phi^{(k)}(\boldsymbol{x}_s^{(i)})\right]_{i=1}^{N_s}, \quad \boldsymbol{Z}_q^{(k)} = \left[\boldsymbol{E}_\Phi^{(k)}(\boldsymbol{x}_q^{(i)})\right]_{i=1}^{N_q},$$

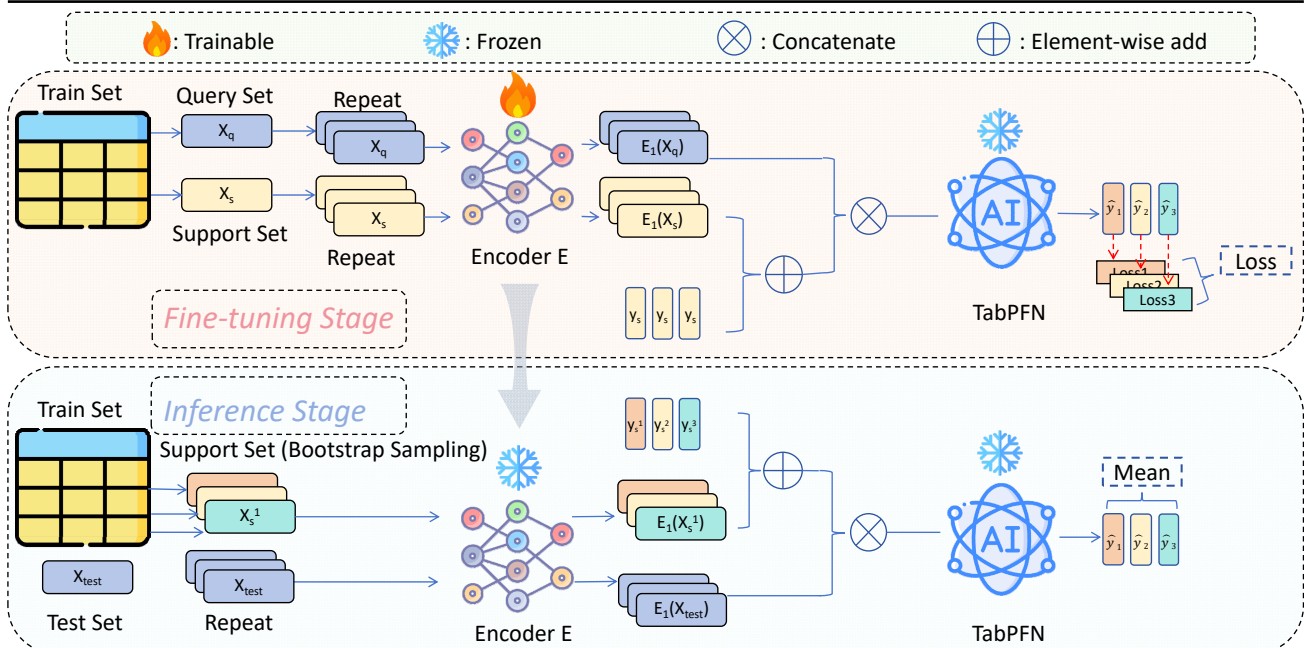

Figure 3: Overview of the proposed method, BETA, which consists of the *fine-tuning stage* and the *inference stage*. During the fine-tuning stage, only the encoder parameters equipped with BatchEnsemble are updated, while the parameters of the original pretrained model remain frozen.

where $N_s$ and $N_q$ denote the number of samples in the support set and the query set, respectively. The multiple encoders are trained jointly by minimizing the sum of the individual losses across all encoders. The optimization objective for the fine-tuning phase is:

$$\min_{\Phi} \mathcal{L}_{\text{total}} = -\sum_{k=1}^{K} \log \left( q_\theta \left( \boldsymbol{y}_q \mid \boldsymbol{Z}_{\text{q}}^{(k)}, \left( \boldsymbol{Z}_{\text{s}}^{(k)}, \boldsymbol{y}_s \right) \right) \right).$$
(3)

Minimizing $\mathcal{L}_{\text{total}}$ in Equation 3 jointly trains all encoders to generate TabPFN-compatible latent representations, with the different encoder initializations encouraging diverse representations and thereby reducing variance and providing more stable predictions.

**Enhancing Performance with Batch Ensemble for Computational Efficiency.** To further enhance performance without introducing additional computational cost, we integrate the Batch Ensemble technique into the encoder. Specifically, we replace the linear layers in $\boldsymbol{E}_{\Phi}$ with Batch Ensemble versions, allowing the model to maintain diversity while avoiding the overhead of training multiple independent encoders. This technique introduces shared weight matrices and member-specific scaling factors, reducing the number of trainable parameters while preserving the benefits of ensembling.

The output of the $k$-th base model for encoder layer $l$ is given by:

$$l_k(\boldsymbol{x}) = s_k \odot (W(r_k \odot \boldsymbol{x})) + b_k$$

where $W$ is shared across all base models, and $r_k, s_k, b_k$ are specific to each base model (Wen et al., 2020).

**Bootstrapped Sampling for Variance Reduction in *Inference Stage*.** To further reduce variance during inference, we apply bootstrapped sampling, generating random subsets of the training set as support sets. These are used to compute predictions across multiple encoders, and their aggregation stabilizes the final output without additional computational overhead. For each encoder $k$, the bootstrapped support set is $D_{\text{bootstrap}}^{(k)} = \left( \boldsymbol{X}_{\text{bootstrap}}^{(k)}, \boldsymbol{Y}_{\text{bootstrap}}^{(k)} \right)$. The final prediction is obtained by averaging over all encoders:

$$\hat{y} = \frac{1}{K} \sum_{k=1}^{K} q_\theta \left( \boldsymbol{E}_{\Phi}^{(k)}(\boldsymbol{x}_{\text{test}}), \left( \boldsymbol{E}_{\Phi}^{(k)}(\boldsymbol{X}_{\text{bootstrap}}^{(k)}), \boldsymbol{Y}_{\text{bootstrap}}^{(k)} \right) \right).$$

This approach reduces variance while maintaining computational efficiency, ensuring scalability and robustness.

**Expanding to MultiClass Tasks Beyond 10 Classes.** To address TabPFN's limitation in handling tasks with more than 10 classes, we integrate the Error-Correcting Output Code (ECOC) (Dietterich & Bakiri, 1995) strategy. Instead of training separate classifiers, distinct encoders within a single model handle individual binary classification tasks, enabling efficient multiclass classification without additional computational overhead.

**Summary of Our Approach.** Our method reduces bias using a lightweight encoder to align input representations with the pre-trained TabPFN, which is effective for high-

dimensional data. Variance is reduced through inference with bootstrapped sampling, enabling better generalization on large datasets. For multiclass tasks, we integrate Error-Correcting Output Codes (ECOC) to efficiently handle more than ten classes. Additionally, we preserve PFN-style batching, ensuring inference efficiency and scalability for large-scale and high-dimensional datasets.

# 4. Experiments

## 4.1. Experiment Setup

**Datasets.** In our experiments, we evaluate BETA on one of the largest publicly available tabular benchmark TALENT (Ye et al., 2024b), which includes 120 binary classification datasets and 80 multi-class classification datasets. These datasets are collected from various sources such as UCI, OpenML, Kaggle, and others. To ensure fairness, we remove two datasets, "PizzaCutter3" and "PieChart3," as they overlap with TabPFN's validation set (Hollmann et al., 2023). In addition, to validate the ability of BETA to handle high-dimensional feature datasets, we also include 20 high-dimensional datasets sourced from the scikit-feature repository.

**Evaluation.** For the TALENT datasets, we follow the evaluation protocol from Gorishniy et al. (2021) and Ye et al. (2024b). Each dataset is randomly split into training, validation, and test sets with proportions of 64%, 16%, and 20%, respectively. For each dataset, we train each model using 15 different random seeds and calculate the average performance on the test set. We report accuracy as the evaluation metric, where higher accuracy indicates better performance. For a more detailed comparison with other TabPFN variants, we separately evaluate datasets with fewer than 10 classes, those with more than 10 classes, and high-dimensional datasets. Further details on the experimental setup, are provided in Appendix B.

**Methods Compared.** We compare BETA against five categories of methods to evaluate its effectiveness comprehensively: (1) **Classical Machine Learning Algorithms:** This category includes widely used classical approaches such as Support Vector Machines (SVM), K-Nearest Neighbors (KNN), and tree-based methods like Random Forest (RForest) (Breiman, 2001), XGBoost (XGB) (Chen & Guestrin, 2016), CatBoost (CatB) (Prokhorenkova et al., 2018), and LightGBM (LightG) (Ke et al., 2017). (2) **Tabular Deep Learning Models:** We consider state-of-the-art deep learning models for tabular data, including MLP, ResNet, FT-Transformer (FT-T) (Gorishniy et al., 2021), MLP-PLR (Gorishniy et al., 2022), DCNv2 (Wang et al., 2021), AutoInt (Song et al., 2019), SNN (Klambauer et al., 2017), ExcelFormer (ExcelF) (Chen et al., 2024), DANets (Chen et al., 2022), TabTransformer

(TabT) (Huang et al., 2020), and TabNet (Arik & Pfister, 2021). (3) **Neighborhood-Based Methods:** To explore neighbor-based strategies, we evaluate TabR (Gorishniy et al., 2024) and ModernNCA (MNCA) (Ye et al., 2025b). (4) **Ensemble-Based Methods:** We include methods that leverage ensemble strategies, such as NODE (Popov et al., 2020), GrowNet (Badirli et al., 2020), and TabM (Gorishniy et al., 2025). For TabM, we evaluate two configurations: TabM-32, which uses the setting of 32 base models as in the original paper, and TabM-16, a variant with 16 base models to facilitate a fairer comparison with BETA, which also uses 16 base models by default. (5) **TabPFN and Its Variants:** Lastly, we compare with TabPFN and its recent variants, including KnnPFN (Nagler, 2023; Thomas et al., 2024), LocalPFN (Thomas et al., 2024), MixturePFN (Xu et al., 2025), and TuneTables (Feuer et al., 2024).

**Implementation Details.** All datasets are pre-processed following the methodology outlined in Gorishniy et al. (2021). For deep learning-based methods, we set the batch size to 1024. Hyper-parameters for the compared methods are tuned using Optuna (Akiba et al., 2019), performing over 100 trials. The search ranges for hyper-parameters are determined based on Liu et al. (2024) and the official implementations of each method. Once the optimal hyper-parameters are identified, they are fixed for the final evaluation using 15 random seeds. To ensure a fair comparison, **all TabPFN variants, including BETA**, are evaluated using their **default hyper-parameters without additional tuning**. More details can be found in Appendix B.

## 4.2. BETA: State-of-the-Art Performance

We conducted pairwise significance testing using the Wilcoxon-Holm test (Demsar, 2006) among BETA and all the compared methods. To ensure a fair comparison, we selected 186 datasets from TALENT with fewer than 10 classes, as TabPFN and its variants are not capable of handling datasets with more than 10 classes. From Figure 4, it is evident that BETA outperforms other methods, **even without hyper-parameter tuning**. This includes methods based on nearest neighbors such as TabR and ModernNCA, ensemble-based approaches like TabM, traditional tree models, and other TabPFN variants. These results underscore the significant potential of pre-trained models for tabular data and demonstrate the effectiveness of our proposed method in adapting to downstream datasets.

**Scale to Large Datasets.** We evaluated BETA and the variants of TabPFN on 173 tabular datasets, excluding those with more than 100 features, which TabPFN cannot handle. These datasets were sorted by the number of rows in ascending order, and the relative improvement of these methods over TabPFN was shown in Figure 5. The results clearly show that the performance improvement

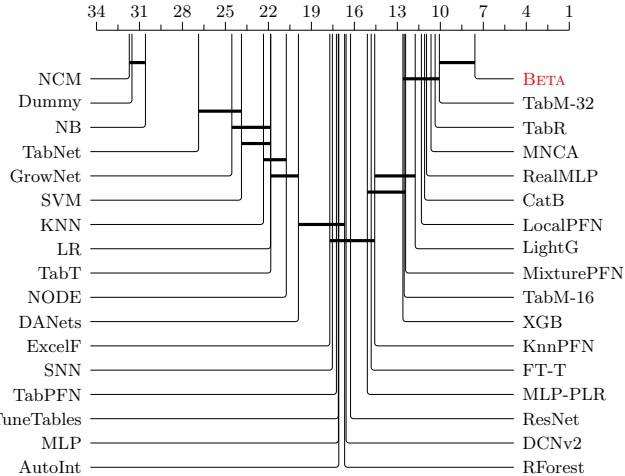

Figure 4: The critical difference diagrams based on the Wilcoxon-Holm test with a significance level of 0.05 to detect pairwise significance for TALENT datasets with fewer than 10 classes. The correspondence between method names and their abbreviations can be found in (Ye et al., 2024b).

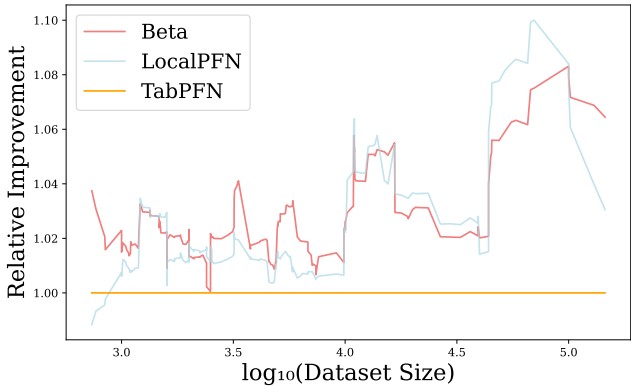

Figure 5: The relative improvement of LocalPFN (Thomas et al., 2024) and BETA (Ours) over TabPFN across 173 tabular datasets. The x-axis represents the logarithm (base 10) of dataset size (number of rows). The curves represent smoothed results to better illustrate trends in performance improvement as dataset size increases. For clarity of presentation, the curves have been smoothed.

by BETA becomes more pronounced as the dataset size increases, highlighting the scalability and effectiveness of our approach in handling larger datasets. Notably, on the largest datasets in the benchmark, LocalPFN slightly outperforms BETA in terms of accuracy. However, on more other datasets, BETA demonstrates superior performance.

**Handling High-Dimensional Datasets.** To assess the effectiveness of BETA on high-dimensional datasets, we conducted experiments on 20 datasets with extremely high feature dimensions, as detailed in Table 2. These datasets were selected to evaluate the scalability and adaptability of different methods in complex feature spaces. The average ranks of the compared methods are summarized in Figure 7. The results show that BETA achieves the best performance, attaining the lowest average rank and outperforming all other methods. TabM and MLP also demonstrate competitive results, ranking second and third, respectively. In contrast, traditional models such as RandomForest and XGBoost, as well as deep learning-based ModernNCA, exhibit lower ranks, highlighting their limitations in high-dimensional settings. Due to memory constraints, we were unable to compare with methods such as FT-T and TabR in this setting.

**Performance on MultiClass Classification Tasks with More Than 10 Classes.** In addition, we further investigated the performance of BETA on classification tasks with more than 10 categories. We selected 12 classification datasets with more than 10 classes from TALENT (Ye et al., 2024b) and compared BETA with other methods. We report the mean accuracy of each method across the 12 datasets in Figure 8 and find that BETA outperforms the compared methods, demonstrating its superiority on multiclass classi-

fication tasks with a larger number of categories.

### 4.3. Ablation Study

We perform an ablation study on the classification datasets of TinyBench2 (Ye et al., 2024b) to examine the contributions of key components, including the number of encoder paths, periodic activation, Batch Ensemble, and partial parameter tuning. Our results indicate that each of these design elements plays a crucial role in enhancing BETA's performance.

**Influence of Encoder Path Count on Performance Improvement over TabPFN.** Figure 6 (a) presents the results of an ablation experiment designed to investigate the impact of different encoder path counts on the performance of BETA. The primary goal is to explore how the number of encoder paths influences the model's relative improvement over TabPFN. We display the relative improvement for BETA with different encoder path configurations. From the plot, we observe that as the number of encoder paths (denoted by $K$) increases, the relative performance improvement of BETA over TabPFN becomes more pronounced. This trend indicates that a higher number of encoder paths contributes positively to the model's ability to outperform TabPFN, suggesting that more encoder paths enhance the model's expressive power and its ability to capture more complex patterns in the task.

**Impact of Some Key Components.** Figure 6 (b) visualizes the effect of three key components—periodic activation, Batch Ensemble, and partial parameter tuning—on the relative performance improvement of BETA over TabPFN. In this experiment, we compare the original BETA with three

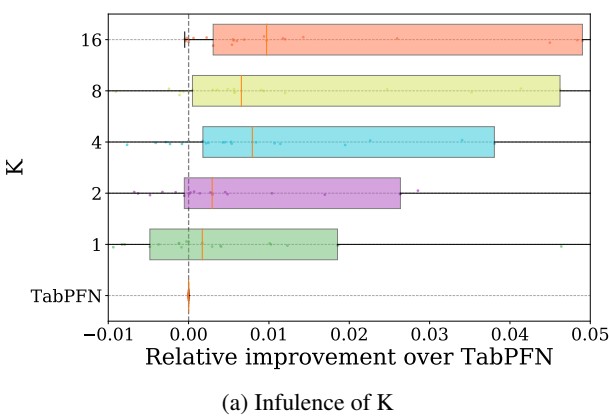

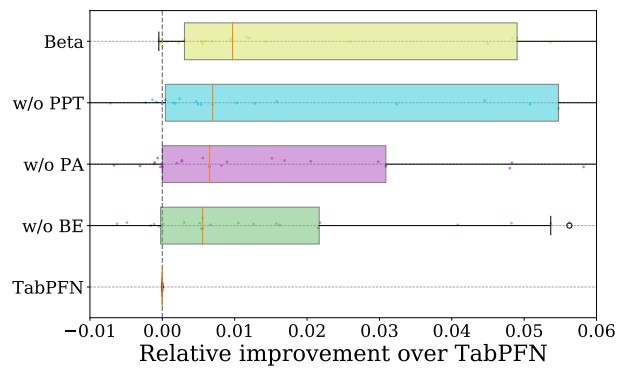

(a) Infulence of K

(b) Influence of some important components

Figure 6: (a) The box plot demonstrates the effect of varying the number of encoder paths, $K$, on the relative improvement over TabPFN. The horizontal axis represents the relative improvement in performance, while the vertical axis lists numbers of different encoder path settings; (b) This box plot illustrates the effect of different components—periodic activation (PA), Batch Ensemble (BE), and partial parameter tuning (PPT)—on the relative improvement of BETA over TabPFN. The plot compares the original BETA with three variations: (1) BETA with partial parameter tuning (w/o PPT), (2) BETA without periodic activation (w/o PA), and (3) BETA without Batch Ensemble (w/o BE).

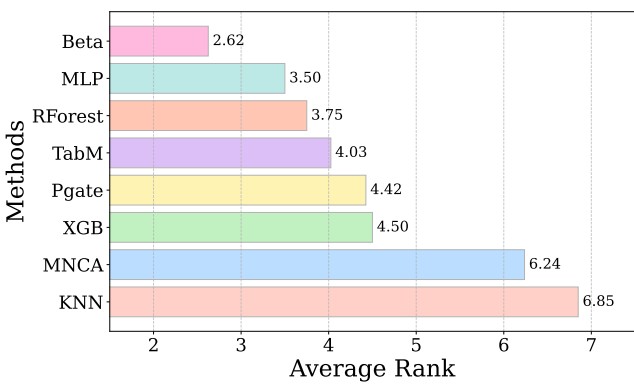

Figure 7: Average ranks of methods on 20 high-dimensional datasets. We compare BETA with TabM, KNN, MLP, XGBoost (XGB), RandomForest (RForest), ProtoGate (Pgate) (Jiang et al., 2024b), and ModernNCA (MNCA). Lower ranks indicate better performance.

variations: one without periodic activation, one without Batch Ensemble, and one with full parameter tuning. The plot reveals that all of these components contribute positively to the model's performance. Specifically, removing any of these elements leads to a noticeable decline in relative improvement over TabPFN, highlighting that all parts are essential for achieving the best performance. Notably, the variation without partial parameter tuning (PPT) shows the highest upper-bound improvement, but it also incurs significantly higher computational costs. Moreover, the median relative improvement decreases compared to BETA, further emphasizing the superior efficiency of our approach. This suggests that periodic activation, Batch Ensemble, and partial parameter tuning each play a crucial role in enhancing

the model's ability to outperform TabPFN, and the absence of any of these components compromises the model's ability to leverage its full potential while also increasing computational resource usage. For more comprehensive and detailed ablation experiments, please refer to Appendix D.5.

## 5. Conclusion

In this paper, we propose BETA, a novel approach that enhances the performance of TabPFN by simultaneously addressing both bias and variance. Through a combination of lightweight encoder-based fine-tuning and bootstrapped sampling, BETA significantly improves TabPFN's adaptability to high-dimensional, large-scale, and multiclass classification tasks. Our method efficiently reduces bias by aligning downstream data distributions with the pre-trained TabPFN and reduces variance through diverse latent representations and robust inference techniques. Experimental results on over 200 benchmark datasets demonstrate that BETA consistently outperforms or matches state-of-the-art methods, highlighting its potential to handle complex tabular data tasks with enhanced scalability and robustness. These contributions provide a scalable and effective solution for leveraging TabPFN in real-world applications, ensuring its success across a broad range of tabular data challenges.

## Acknowledgements

This work is partially supported by National Key R&D Program of China (2024YFE0202800), NSFC (62376118), Key Program of Jiangsu Science Foundation (BK20243012), Collaborative Innovation Center of Novel Software Technology and Industrialization.

## Impact Statement

Our work advances the field of tabular learning and in-context learning (ICL) by introducing a method that significantly improves the scalability, adaptability, and performance of TabPFN. Given that tabular data is widely used across industries such as finance, healthcare, and e-commerce, achieving state-of-the-art classification accuracy has far-reaching benefits. Our approach enhances the usability of tabular foundation models (TFMs), providing a more effective and scalable alternative to conventional methods while preserving inference efficiency. We hope that our results inspire further research into ICL-driven approaches for tabular data and their applications in broader real-world tasks. As our method is built on pre-trained TabPFN, its deployment in high-stakes applications should consider potential limitations, such as reliance on the model's prior and sensitivity to dataset characteristics. We do not foresee significant negative societal impacts arising from our work.

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

The Appendix consists of four sections:

# A. Related Work

### A.1. Tabular Data Learning

Tabular data is one of the most widely used dataset types in machine learning. Gradient-boosted decision trees (GBDTs) (Chen & Guestrin, 2016; Prokhorenkova et al., 2018; Ke et al., 2017), remain a strong baseline for tabular tasks due to their efficiency and high performance. As an ensemble-based method, GBDTs construct multiple decision trees to iteratively minimize the residual loss. With the advancement of deep learning, an increasing number of studies have explored using deep learning methods for tabular data prediction (Borisov et al., 2024a; Jiang et al., 2025). These approaches include MLP variants (Klambauer et al., 2017; Gorishniy et al., 2021; 2022; Holzmüller et al., 2024), neural networks specifically designed for tabular structures (Wang et al., 2017; 2021; Chen et al., 2023a), attention-based models (Song et al., 2019; Huang et al., 2020; Gorishniy et al., 2021; Chen et al., 2024; Jiang et al., 2024a; Zhou et al., 2023), methods incorporating regularization (Ye et al., 2024a; Alan et al., 2023; Wu et al., 2024), tree-mimic methods (Arik & Pfister, 2021; Popov et al., 2020; Badirli et al., 2020), and context-based methods (Gorishniy et al., 2024; Ye et al., 2025b). Despite these advancements, recent benchmarks (Grinsztajn et al., 2022; McElfresh et al., 2023; Ye et al., 2024b) have consistently demonstrated that gradient-boosted decision trees (GBDTs) outperform deep learning in tabular prediction tasks. The superior performance of GBDTs can be attributed to two main factors: (1) their ability to handle heterogeneous tabular datasets, which often describe high-frequency target functions (Basri et al., 2020; Grinsztajn et al., 2022), and (2) the ensemble nature of GBDTs. Prior attempts to introduce ensemble-like mechanisms into tabular deep learning (Badirli et al., 2020; Popov et al., 2020; Chen et al., 2023c), have not been widely successful (Grinsztajn et al., 2022; Ye et al., 2024b). However, recent works like TabM (Gorishniy et al., 2025) integrate Batch Ensemble (Wen et al., 2020) techniques into the tabular domain, showing how efficient ensembling can be achieved with deep learning models.

### A.2. Tabular Foundation Models

While tabular foundation models (TFMs) are not as developed as foundation models in other domains, such as computer vision (Saharia et al., 2022) and natural language processing (Brown et al., 2020), recent efforts have introduced various architectures to bridge this gap (van Breugel & van der Schaar, 2024). Some approaches aim to explore model components that can be shared across datasets (Liu et al., 2022; Zhu et al., 2023), while others focus on utilizing the semantic information inherent in tabular datasets (Wang & Sun, 2022; Yan et al., 2024; Gardner et al., 2024; Yang et al., 2024; Kim et al., 2024; Kasneci & Kasneci, 2024; Liu et al., 2025; Koloski et al., 2025). Transformer-based models, such as TabPFN (Hollmann et al., 2023; 2025; Ye et al., 2025a) and TabICL (Qu et al., 2025) stand out for their exceptional performance and efficiency on small datasets. By leveraging the in-context learning capability of transformer (Brown et al., 2020), they can make predictions for unseen instances without parameter updates. However, TabPFN faces limitations related to dataset size and feature dimensionality. To address these challenges, some studies have explored improvements in its architecture and training paradigm (den Breejen et al., 2024; Ma et al., 2024b), while others focus on adaptation techniques that expand TabPFN's applicability to a broader range of downstream tabular datasets (Feuer et al., 2024; Thomas et al., 2024; Koshil et al., 2024). Our work falls into the latter category, offering a straightforward, efficient, and effective approach to align TabPFN with downstream datasets.

### A.3. Parameter-efficient Fine-Tuning

Some approaches for adapting Tabular Foundation Models (TFMs) to downstream tasks simply fine-tune the entire model (Thomas et al., 2024; den Breejen et al., 2024), which leads to performance improvements but incurs high computational and storage costs. Parameter-efficient fine-tuning (PEFT) offers a solution to the challenge by enabling adaptation with a minimal number of trainable parameters (Lialin et al., 2023). Based on their operational mechanisms, Han et al. (2024) broadly categorized PEFT methods into four paradigms:

- **Additive Methods**. These methods incorporate additional lightweight modules into the model architecture, such as adapters (Houlsby et al., 2019; Zhang et al., 2020; Xu et al., 2025) or soft prompts (Li & Liang, 2021; Feuer et al., 2024).
- **Selective Methods**. Instead of introducing new parameters, selective methods strategically identify and update the most relevant parameters while freezing the rest (Fu et al., 2023; He et al., 2023).
- **Reparameterized Methods**. These methods employ low-rank decomposition or equivalent transformations to reduce the parameter space during fine-tuning (Aghajanyan et al., 2021; Hu et al., 2022; Valipour et al., 2023).
- **Hybrid Methods**. By combining the strengths of multiple PEFT strategies, hybrid methods create a unified framework that enhances fine-tuning performance while maintaining efficiency (Mao et al., 2022; Chen et al., 2023b).

Our proposed method adopts parameter-efficient tuning focused on input feature adaptation. This design is motivated by the unique characteristics of tabular data: Tabular datasets are inherently heterogeneous, with varying structures and feature distributions across different datasets (Zhou et al., 2023; Ye et al., 2023; Borisov et al., 2024b). Adapting TabPFN through input feature alignment effectively mitigates the constraints on input dimensionality, enhancing its applicability across a wider range of tasks. By employing parameter-efficient fine-tuning, we align TabPFN with downstream datasets, addressing its existing limitations (Hollmann et al., 2023).

## B. Datasets and implementation details

In this section, we outline the descriptions of the datasets used in the experiments and the preprocessing steps applied to them before training. Additionally, we will describe the implementation details of BETA and the comparison methods.

### B.1. Datasets information

For both the main experiments and the classification tasks with more than 10 categories, we use the current largest tabular benchmark (Ye et al., 2024b), which includes **120 binary classification datasets and 80 multiclass classification** datasets, spanning diverse domains such as healthcare, biology, finance, education, and physics. For more detailed information about these datasets, please refer to Ye et al. (2024b).

For each dataset, we randomly select 20% of the instances to form the test set. The remaining 80% is further split, with 20% reserved as a validation set. This validation set is used for hyperparameter tuning (for the comparison methods) and early stopping. The hyperparameters that yield the best performance on the validation set are selected for final evaluation on the test set.

*Remark*: BETA are based on pre-trained models TabPFN (Hollmann et al., 2023). To ensure a fair comparison, we do not perform hyperparameter search, and instead use the default hyperparameters for all TabPFN-based methods.

For high-dimensional datasets, we obtained the data from scikit-feature repository, and excluded those with more than 10 classes. This resulted in a final set of 20 datasets, as shown in Table 2. The dataset splitting follows the same approach as in the main experiments, but due to the smaller number of instances in these datasets, we use the default hyperparameters for the experiments and do not perform hyperparameter search. The validation set is only used for early stopping. For each dataset, we perform five random splits and run all methods three times on each split using three different seeds (0, 1, 2), resulting in a total of 15 runs per dataset. The final results are reported as the mean of these 15 runs.

### B.2. Dataset Pre-processing

Unless otherwise specified, we adopt the data preprocessing pipeline outlined by Gorishniy et al. (2021). For numerical features, we standardize by subtracting the mean and scaling the values to unit variance. Categorical features are converted into a model-compatible format using one-hot encoding.

### B.3. Implementations

BETA. For all experiments except for the ablation study, we set the context size to 1000 and fixed the number of bootstrap sampling iterations to 16. For the feature transformation encoder, the main structure is a two-layer MLP, with both the hidden and output dimensions set to 100, using the ReLU activation function. In addition, to enhance the expressive power of the encoder, we also apply periodic activation (Gorishniy et al., 2022). The initialization of Batch Ensemble is inspired by Gorishniy et al. (2025). During fine-tuning, we use the pre-trained model checkpoint[1], and fine-tuning is performed

---

[1]https://github.com/automl/TabPFN/blob/tabpfn_v1/tabpfn/models_diff/prior_diff_real_checkpoint_n_0_epoch_42.cpkt

Table 2: Dataset Information for High-Dimensional Data Experiments: A collection of 20 datasets with varying numbers of instances, features, and classes used in our high-dimensional experiments.

| Dataset | #Instances | #Features | #Classes | Dataset | #Instances | #Features | #Classes |
|---------|-----------|-----------|----------|---------|-----------|-----------|----------|
| BASEHOCK | 1993 | 4862 | 2 | lung_discrete | 73 | 325 | 7 |
| PCMAC | 1943 | 3289 | 2 | warpPIE10P | 210 | 2420 | 10 |
| RELATHE | 1427 | 4322 | 2 | orlraws10P | 100 | 10304 | 10 |
| ALLAML | 72 | 7129 | 2 | Prostate_GE | 102 | 5966 | 2 |
| CLL_SUB_111 | 111 | 11340 | 3 | SMK_CAN_187 | 187 | 19993 | 2 |
| colon | 62 | 2000 | 2 | warpAR10P | 130 | 2400 | 10 |
| GLI_85 | 85 | 22283 | 2 | arcene | 200 | 10000 | 2 |
| GLIOMA | 50 | 4434 | 4 | gisette | 7000 | 5000 | 2 |
| leukemia | 72 | 7070 | 2 | madelon | 2600 | 500 | 2 |
| lung | 203 | 3312 | 5 | TOX_171 | 171 | 5748 | 4 |

using the AdamW (Loshchilov & Hutter, 2019) optimizer with a learning rate of 0.003, weight decay of 1e-5, and a batch size of 1024. In the fine-tuning phase, only the encoder parameters are updated, while the pre-trained TabPFN parameters are frozen. For experiments on high-dimensional datasets, due to the large number of features and the limitations of device memory, we do not use periodic activation. In the classification experiments with more than 10 classes, we use a code length of 32, which means there are 32 paths in the encoder. Additionally, we observed that for certain datasets, simply adjusting the output dimension of the pre-trained model and fine-tuning it can yield strong results (Feuer et al., 2024). Based on validation set performance, we choose between ECOC and fine-tuning the output layer MLP approaches.

**TabPFN and its variants.** For TabPFN, we use the implementation from TALENT (Liu et al., 2024); For TuneTables, we adopt the official code[2] and use the prompt tuning mode without performing full model fine-tuning; For LocalPFN, since the code has not been released, we replicate the hyperparameters provided in the original paper. Specifically, we set the number of neighbors $k = 1000$. During our experiments, we found that fine-tuning LocalPFN with a smaller learning rate yields better performance, so we set the learning rate to 1e-5; For MixturePFN, we reproduce the model using the hyperparameters recommended in the original paper. The number of training samples in the prompt is set to $B = 3000$, and the number of experts is set to the training set size divided by $B$, rounded up to the nearest integer.

**Other Methods from TALENT.** For the other methods in TALENT, we either use the results provided (Ye et al., 2024b) or perform a hyperparameter search using the hyperparameters provided by TALENT. In the case of hyperparameter tuning, we conduct 100 trials of hyperparameter search for each method[3]. For each search, we run the experiments using 15 different seeds and report the average results across these runs.

### B.4. Details of TabPFN

In the original TabPFN approach, the model processes the input data in two phases: pre-training and inference. The pre-training phase involves training the model on synthetic data, while the inference phase is used to make predictions on new, unseen data.

For the input, we consider a training dataset $D_{\text{train}} = (X_{\text{train}}, y_{\text{train}})$, where $X_{\text{train}} \in \mathbb{R}^{N_{\text{train}} \times d}$ represents the feature matrix, and $y_{\text{train}} \in \mathbb{R}^{N_{\text{train}}}$ represents the corresponding labels. Here, $N_{\text{train}}$ denotes the number of training samples, and $d$ is the number of features. For datasets with fewer features than the fixed input dimensionality $d_{\max}$, zero-padding is applied to extend the feature vectors to the required size $d_{\max}$. Specifically, each feature vector $X_i \in \mathbb{R}^d$ is padded with zeros to form $\tilde{X}_i \in \mathbb{R}^{d_{\max}}$, as follows:

$$\tilde{X}_i = [X_i, \mathbf{0}] \in \mathbb{R}^{d_{\max}}.$$

The zero-padded feature matrix $\tilde{X}_{\text{train}}$ is then used during inference for making predictions.

---

[2]https://github.com/penfever/TuneTables
[3]https://github.com/qile2000/LAMDA-TALENT/tree/main/LAMDA_TALENT/configs

For inference, we consider a support set $S$ containing the training examples and a query set $Q$ containing the new, unseen instances for which we wish to make predictions. The query set $Q$ consists of $N_{\text{test}}$ test samples, and the goal is to predict the corresponding labels $y_{\text{test}}$.

TabPFN uses **PFN-style batching** for inference, where both the support set $S$ and query set $Q$ are batched together into a single prompt, which is then fed into the pre-trained model for prediction. This batching process allows the model to utilize the support set as context for making predictions on the query set.

During the processing, the features of both the support set and the query set are transformed into token representations. The support tokens $L_{\text{support}}$ are obtained as:

$$L_{\text{support}} = \tilde{X}_{\text{support}} W_x + y_{\text{support}} w_y^T,$$

where $\tilde{X}_{\text{support}} \in \mathbb{R}^{|S| \times d_{\max}}$ is the zero-padded feature matrix of the support set, $W_x \in \mathbb{R}^{d_{\max} \times d_{\text{token}}}$ is the embedding matrix for the features, and $w_y \in \mathbb{R}^{d_{\text{token}}}$ is the embedding matrix for the labels. The query tokens $L_{\text{query}}$ are similarly transformed as:

$$L_{\text{query}} = X_{\text{query}} W_x,$$

where $X_{\text{query}} \in \mathbb{R}^{|Q| \times d_{\max}}$ represents the query set of test samples.

These token representations are then passed through a standard transformer model, which includes a special attention mechanism. The attention mask ensures that the support tokens can only attend to other support tokens, and query tokens can attend to both the support tokens and themselves. Query tokens do not attend to other query tokens, preventing any information leakage between query samples during inference. Finally, the output tokens corresponding to the test instances are extracted and mapped to 10-class logits for classification.

### B.5. Implementation of TabPFN-Bagging

TabPFN-Bagging is a variance reduction technique that leverages **bootstrapped sampling** to create diverse support sets for model inference. Unlike standard ensemble approaches, which require training multiple independent models, TabPFN-Bagging generates multiple resampled versions of $D_{\text{train}}$ within a single inference process, reducing variance without additional computational overhead.

**Bootstrapped Sampling in TabPFN.** Given a training dataset $D_{\text{train}} = \{(\boldsymbol{x}_i, \boldsymbol{y}_i)\}_{i=1}^{N_{\text{train}}}$, we generate $K$ bootstrapped support sets $D_{\text{train}}^{(k)}$, where each set is constructed by randomly sampling $N_{\text{sub}}$ instances with replacement:

$$D_{\text{train}}^{(k)} = \left\{ \left( \boldsymbol{x}_i^{(k)}, \boldsymbol{y}_i^{(k)} \right) \right\}_{i=1}^{N_{\text{sub}}}, \quad k = 1, \dots, K.$$

Each $D_{\text{train}}^{(k)}$ serves as an alternative context set for model inference, introducing diversity into the prediction process.

**Prediction Aggregation.** For each test instance $\boldsymbol{x}_{\text{q}}$, we compute the posterior predictive distribution using the bootstrapped support sets. Given the pre-trained TabPFN model parameterized by $\theta$, the predictive probability for label $\boldsymbol{y}_{\text{q}}$ is computed as:

$$p_\theta^{(k)}(\boldsymbol{y}_{\text{q}} \mid \boldsymbol{x}_{\text{q}}, D_{\text{train}}^{(k)}) = \frac{\exp(q_\theta(\boldsymbol{x}_{\text{q}}, D_{\text{train}}^{(k)})_{[\boldsymbol{y}_{\text{q}}]})}{\sum_{c=1}^{C} \exp(q_\theta(\boldsymbol{x}_{\text{q}}, D_{\text{train}}^{(k)})_{[c]})}.$$

where $q_\theta$ denotes the logits produced by TabPFN for the given support set $D_{\text{train}}^{(k)}$.

To obtain the final prediction, we aggregate the results across all bootstrapped support sets using either uniform averaging:

$$p_\theta(\boldsymbol{y}_{\text{q}} \mid \boldsymbol{x}_{\text{q}}, D_{\text{train}}) = \frac{1}{K} \sum_{k=1}^{K} p_\theta^{(k)}(\boldsymbol{y}_{\text{q}} \mid \boldsymbol{x}_{\text{q}}, D_{\text{train}}^{(k)}),$$

or a weighted aggregation method, where weights $w_k$ are assigned based on the confidence of each individual model:

$$p_\theta(\boldsymbol{y}_{\text{q}} \mid \boldsymbol{x}_{\text{q}}, D_{\text{train}}) = \sum_{k=1}^{K} w_k p_\theta^{(k)}(\boldsymbol{y}_{\text{q}} \mid \boldsymbol{x}_{\text{q}}, D_{\text{train}}^{(k)}), \quad \sum_{k=1}^{K} w_k = 1.$$

**Computational Efficiency.** Since bootstrapped sampling only modifies the selection of $D_{\text{train}}$ without altering the model architecture, TabPFN-Bagging incurs no additional computational cost beyond standard inference with ensemble. Unlike ensemble-based methods that require multiple independent model evaluations, all computations occur within a single forward pass of the pre-trained TabPFN. This makes TabPFN-Bagging an efficient and scalable variance reduction strategy.

## C. Hardware and limitations

### C.1. Hardware

Most of the experiments were performed with four NVIDIA GeForce RTX 4090 GPUs, four NVIDIA RTX 6000 Ada GPUs, and eight NVIDIA GeForce RTX 3090 GPUs.

### C.2. Limitations

While our approach significantly improves the scalability and adaptability of TabPFN, it has certain limitations that present opportunities for future research. Due to the current design of TabPFN, our method has been evaluated solely on classification tasks, and extending it to regression tasks remains an open challenge. As tabular foundation models continue to evolve, we anticipate that our approach can be adapted to support regression, broadening its applicability to a wider range of real-world problems.

## D. Additional Experiments

### D.1. Performance on MultiClass Classification Tasks with More Than 10 Classes

To evaluate the effectiveness of BETA on multiclass classification tasks, we conducted experiments on 12 datasets containing more than 10 classes, sourced from TALENT(Ye et al., 2024b). These datasets were selected to assess the scalability and adaptability of different methods in handling more complex classification problems beyond the 10-class limitation of TabPFN. Figure 8 presents the mean accuracy of each method across these datasets. We compare BETA against several strong baselines, including KNN, XGBoost (Chen & Guestrin, 2016), RandomForest (Breiman, 2001), MLP (Gorishniy et al., 2021), ModernNCA (Ye et al., 2025b), and TabM (Gorishniy et al., 2025). All methods, including BETA, were evaluated using their default hyperparameters without any tuning, ensuring a fair comparison of out-of-the-box performance. From the results, BETA achieves the highest mean accuracy across all datasets, demonstrating superior generalization ability

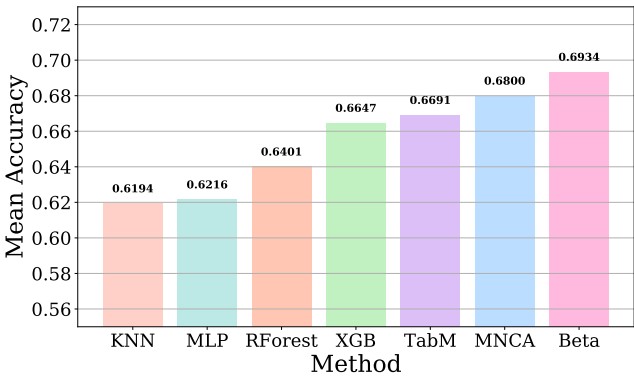

Figure 8: Comparison of the mean accuracy across 12 classification datasets with more than 10 classes. **Since the original TabPFN cannot process classification tasks with more than ten categories**, we compare BETA against alternative methods, including TabM, KNN, MLP, XGBoost (XGB), RandomForest (RForest), and ModernNCA (MNCA).

in high-category classification settings. Notably, tree-based methods like XGBoost and RandomForest exhibit a decline in performance as the number of categories increases (Gorishniy et al., 2021; Ye et al., 2024b).

### D.2. Performance on Real Datasets: Bias and Variance

The generalization error of TabPFN can be attributed to two primary components: **bias** and **variance**. These components are influenced by the model's architecture, pretraining assumptions, and dataset characteristics. Mathematically, the

generalization error can be decomposed by analyzing the discrepancy between the model's predictions, $q_\theta(y \mid x, D_n)$, and the true conditional distribution, $p_0(y \mid x)$. Specifically, it can be expressed as:

$$q_{\boldsymbol{\theta}}\left(y \mid \boldsymbol{x}, \mathcal{D}_n\right) - p_0(y \mid \boldsymbol{x}) = \underbrace{q_{\boldsymbol{\theta}}\left(y \mid \boldsymbol{x}, \mathcal{D}_n\right) - \mathbb{E}_{\mathcal{D}_n \sim p_0^n}\left[q_{\boldsymbol{\theta}}\left(y \mid \boldsymbol{x}, \mathcal{D}_n\right)\right]}_{\text{Variance}} + \underbrace{\mathbb{E}_{\mathcal{D}_n \sim p_0^n}\left[q_{\boldsymbol{\theta}}\left(y \mid \boldsymbol{x}, \mathcal{D}_n\right)\right] - p_0(y \mid \boldsymbol{x})}_{\text{Bias}}. \quad (4)$$

Here, $\mathbb{E}[\cdot]$ denotes the expectation over training datasets $D_n$ sampled from the true distribution $p_0$. Understanding these error components is crucial for addressing the limitations of TabPFN and improving its performance across diverse tabular datasets.

Revisiting the results shown in Figure 2, we observe several key phenomena that highlight the effectiveness of BETA in reducing generalization error across varying dataset sizes:

**Lowest Generalization Error Across All Dataset Sizes:** Our method consistently achieves the smallest generalization error, demonstrating robust performance regardless of dataset size.

**Impact of Fine-Tuning on Small Datasets:** For small datasets (e.g., with only a few hundred samples), fine-tuning the encoder may slightly increase bias. This occurs because the limited data volume can lead to overfitting during fine-tuning.

**Mechanisms of Bias and Variance Reduction:** The bias reduction is primarily attributed to the fine-tuned encoder, which aligns TabPFN with the downstream dataset distribution, improving compatibility between the pre-trained model and the target task. Variance reduction, on the other hand, is not solely due to Bagging. The learning of multiple encoders also contributes significantly. When dataset sizes are small (e.g., fewer than 1000 samples), where each bootstrap sample covers the entire training set, training multiple encoders still introduces diversity in representations, further reducing variance.

These findings highlight the effectiveness of BETA in addressing the bias-variance tradeoff, making it a robust and scalable solution for real-world tabular learning.

To further support our findings, Table 3 presents the relative percentage changes in bias and variance (negative values indicate reduction) of our method Beta and some variants compared to TabPFN. The values are averaged over the dataset sizes {200, 400, 800, 1600, 3200, 6400, 12800} used in Figure 2 on the Adult and Bank datasets. For completeness, we also include results for LocalPFN (Thomas et al., 2024) and a simplified version of BETA, denoted as BETA-S, which uses only a single encoder (*i.e.*, without Batch Ensemble and bagging)., which are not shown in Figure 2. Since TabPFN's absolute variance on these datasets is smaller than its bias, the percentage changes in bias are accordingly smaller.

Table 3: Relative percentage changes (%) in bias and variance compared to TabPFN. Negative values denote reductions.

| Metric | Dataset | BETA | BETA-S | KnnPFN | LocalPFN | TabPFN-Finetune | TabPFN-Bagging |
|--------|---------|------|--------|--------|----------|-----------------|----------------|
| *Bias* | Adult | **-4.40** | -3.60 | -0.77 | -1.68 | -2.69 | +1.73 |
|        | Bank | **-1.65** | -1.19 | -1.33 | -1.47 | -1.05 | +0.53 |
| *Variance* | Adult | **-16.23** | +46.99 | +4.99 | +18.52 | +39.27 | -9.91 |
|            | Bank | **-36.57** | +27.12 | +27.49 | +2.06 | +10.35 | -9.32 |

## D.3. Comparison with Bagging Versions of Standard Baselines and Tabular Foundation Models

To further contextualize the performance of BETA, we conduct a comparative study between recent tabular foundation models and classical machine learning baselines with and without Bagging. Specifically, we include TabPFN v2 (Hollmann et al., 2025) and TabICL (Qu et al., 2025) as representative foundation models. For each, we report two variants: the original version using default hyperparameters as described in their respective papers, and a standardized version using the same context size (1000) and ensemble size (16) as BETA, denoted as TabPFN-v2-S and TabICL-S, respectively.

While we only applied inference-stage bagging for TabPFN in the main experiments, this analysis systematically evaluates the effect of Bagging on a wider set of models. For CatBoost (Prokhorenkova et al., 2018), LightGBM (Ke et al., 2017), and RealMLP (Holzmüller et al., 2024), we trained 16 independent models per dataset using different random seeds. Each model was trained on a subsampled training set of size $\max(0.8 \times \text{train\_size}, 1000)$ to ensure the subset size is comparable to that used by BETA during inference. Notably, TabPFN allows for changing the context size at inference time without retraining. For other baselines, bagging would significantly increase the cost—scaling nearly linearly with the number of base models. We report the average ranks of above methods in Table 4. The results highlight that Bagging substantially improves the

performance of traditional baselines. Results show that the full versions of TabPFN-v2 and TabICL outperform BETA, while their subsampled variants (TabPFN-v2-S and TabICL-S) perform significantly worse. This highlights the sensitivity of these foundation models to their configuration settings, particularly context size and ensemble strategy. Currently, BETAis constrained by memory limitations inherent to our implementation. However, we believe that a well-optimized version of TabPFN-v1 could support larger context sizes with lower memory overhead. This would allow our method to scale more effectively and potentially yield even stronger performance.

| Compared Methods | BETA | TabPFN v2 | TabICL | TabPFN v2-S | TabICL-S |
|---|---|---|---|---|---|
| Average Rank | 4.63 | 4.07 | 4.40 | 5.83 | 5.84 |
| Catboost | Catboost-B | XGBoost | XGBoost-B | RealMLP | RealMLP-B |
| 5.13 | 5.02 | 7.27 | 7.27 | 7.06 | 5.62 |

Table 4: Average ranks of tabular foundation models and classical baselines with and without Bagging. For TabPFN v2 and TabICL, we report both their default configurations and standardized variants (with context size 1000 and ensemble size 16, denoted by "-S"). For traditional methods, we compare vanilla versions against their Bagging counterparts(denoted by "-B"). Lower rank indicates better average performance across datasets.

## D.4. Impact of Cross-Validation on Model Performance

Recent work (Tschalzev et al., 2025) has highlighted methodological concerns in tabular model evaluation protocols, particularly regarding hyperparameter optimization (HPO). One key recommendation is to replace single train/validation splits with cross-validation (CV), to reduce the risk of overfitting and improve the reliability of performance comparisons.

Motivated by these findings, we conducted an experiment to compare standard holdout-based tuning with 5-fold CV on classification datasets with fewer than 3,000 samples. We evaluated three representative methods: XGBoost, RealMLP, and BETA. For XGBoost and RealMLP, we performed HPO using 5-fold CV and ensembled the predictions of the five resulting models. For BETA, which use default hyperparameters, we trained five models using different training folds and averaged their predictions during inference.

Table 5: Comparison of model performance (Average AUC) with and without cross-validation. All results are averaged over the same set of classification datasets with fewer than 3,000 samples.

| Method | Holdout (Avg AUC) | 5-Fold CV (Avg AUC) |
|---|---|---|
| XGBoost | 0.8452 | **0.8470** |
| RealMLP | 0.8610 | **0.8739** |
| BETA | 0.8759 | **0.8785** |

As shown in Table 5, cross-validation consistently improves performance across all three methods. The gain is especially notable for RealMLP, which benefits from more robust tuning. Even for BETA, which uses fixed hyperparameters, averaging across folds yields a modest but measurable improvement.

## D.5. Additional Ablation Study

To show the sensitivity of the performance of BETA to the number of bootstrapped samples and encoder paths, we conduct additional experiments. Table 6 and Table 7 summarize the results on the TinyBench2 classification datasets.

Table 6: Average performance rank of BETA with different numbers of bootstrapped samples across TinyBench2 (Ye et al., 2024b) datasets. Lower is better.

| Bootstrap Size | 500 | 1000 | 2000 | 3000 | TabPFN |
|---|---|---|---|---|---|
| **Average Rank** | 3.39 | 2.87 | 2.57 | 1.91 | 4.24 |

Table 7: Relative improvement (%) of BETA with different numbers of encoders over TabPFN.

| # Encoders | 1 | 2 | 4 | 8 | 16 |
|---|---|---|---|---|---|
| **Relative Improvement** | 1.70% | 2.60% | 3.19% | 3.23% | 3.76% |

These results demonstrate that increasing the number of bootstrapped samples and encoders leads to improved performance, albeit with higher computational costs.

### D.6. Comparison of inference time and the number of learnable parameters.

We compared the inference time (in seconds) of BETA with other methods, as well as the average rank in the main experiment. Additionally, the number of learnable parameters was roughly estimated by the size of the stored checkpoint. The results confirm that BETA not only delivers state-of-the-art classification accuracy but also maintains low inference time and reduced memory overhead, making it highly scalable and efficient for various tabular classification tasks.

Table 8: Comparison of Inference Time, Training Time, and Number of Learnable Parameters. All metrics are computed as the mean values across multiple datasets. For BETA, results are reported with the number of encoders set to 4.

| Metric | BETA | TabPFN | LocalPFN | FT-T | TabR |
|---|---|---|---|---|---|
| **Inference Time (s)** | 0.65 | 0.48 | 12.51 | **0.36** | 0.38 |
| **Checkpoint Size (MB)** | **0.60** | 0 | 98.65 | 7.15 | 14.07 |
| **Finetuning/Training time (s)** | 62.29 | 0 | 186.73 | **58.9** | 121 |

### D.7. Handling Distribution Shifts

To evaluate the robustness of BETA under distribution shift scenarios (e.g., covariate and concept drift), we follow prior work and incorporate plugin techniques into our method.

Specifically, we evaluate BETA on three classification datasets from the TabRed benchmark (Rubachev et al., 2024), which are designed to exhibit temporal shifts. We integrate two techniques discussed in Cai & Ye (2025) into our method:

- **BETA-split**: A temporal-aware splitting method that assigns samples closer to the test set in time to the training set, and those further away to the validation set.

- **BETA-temporal**: A variant of BETA that includes temporal embedding to enhance generalization under time-evolving data.

Table 9 reports the AUC scores of BETA and its variants, along with a standard MLP baseline. The results demonstrate that both plugin techniques improve performance, particularly in settings with stronger temporal distribution shifts. These findings underscore the flexibility of ICL-based models like BETAin handling real-world data drift.

Table 9: AUC results of BETAand variants under temporal distribution shifts on TabRed datasets.

| Dataset | BETA | BETA-split | BETA-temporal | MLP |
|---|---|---|---|---|
| ecom-offers | 0.5692 | 0.6264 | **0.6348** | 0.5866 |
| homesite-insurance | 0.9600 | **0.9602** | 0.9521 | 0.9404 |
| homecredit-default | 0.6114 | **0.6710** | 0.6637 | 0.4730 |

### D.8. Visialzation results

To gain deeper insights into the properties of BETA, we visualize the learned embeddings $E_\Phi(x)$ for BETA, MLP, ModernNCA, and TabR using T-SNE (Van der Maaten & Hinton, 2008). As shown in Figure 9, we present the embeddings of

three datasets (*Bank*, *KDD*, and *Spambase*), comparing different methods as well as the representations learned by different encoder paths within BETA.

All deep tabular methods effectively transform the feature space, making it more conducive for classification compared to raw input features. MLP produces well-separated clusters, grouping same-class samples into distinct, compact regions. In contrast, TabR and ModernNCA form multiple clusters for the same class, positioning similar samples closer to each other while maintaining local class separability. Unlike these methods, BETA does not enforce strict clustering of same-class samples into single compact groups. This behavior aligns with TabPFN's pretraining paradigm, where the model does not require fully separable embeddings to make accurate predictions.

Moreover, we observe that different encoder paths in BETA map raw features into diverse representation spaces, contributing to variance reduction and improved robustness. The variation among encoder paths enhances the diversity of the learned representations, which is a key factor in BETA's superior generalization. The visualization further confirms that BETA preserves the flexibility of the feature space while effectively adapting to downstream classification tasks.

### D.9. Detail Results

Table 10: Average accuracy of different methods on 186 datasets. Each value represents the mean accuracy over 15 random seeds. The dataset ID corresponds to the numbering in Ye et al. (2024b). "XGB," "CatB," "RF," "LightG," "TNet," "FTT," "DCN2," "TabT," "GNet," "MNCA," "M-PLR," "RMLP,""ExcelF," and "PFN" denote "XGBoost," "Catboost," "RandomForest," "LightGBM," "TabNet," "FT-Transformer," "DCNv2," "TabTransformer," "GrowNet," "ModernNCA," "MLP-PLR," "RealMLP," "ExcelFormer," and "TabPFN," respectively. In each cell, if the experiment fails to perform due to being out of memory, we denote it by "-." The original TabPFN cannot handle datasets with more than 10 classes or more than 100 features.

| ID | KNN | SVM | XGB | CatB | RF | LightG | MLP | ResNet | NODE | TNet | DaNet | FTT | AutoInt | DCN2 | SNN | TabT | GNet | TabR | MNCA | M-PLR | RMLP | ExcelF | PFN | TabM-16 | **BETA** |
|---|---|---|---|---|---|---|---|---|---|---|---|---|---|---|---|---|---|---|---|---|---|---|---|---|---|
| 8 | 94.25 | 94.28 | 94.48 | 94.67 | 94.22 | 94.56 | 94.27 | 94.20 | 94.22 | 94.18 | 94.23 | 94.20 | 94.31 | 94.21 | 94.31 | 94.50 | 93.78 | 94.39 | 94.29 | 94.29 | 94.36 | 94.37 | 94.22 | 94.29 | 94.22 |
| 10 | 75.96 | 73.66 | 74.55 | 74.45 | 74.22 | 73.99 | 73.44 | 73.55 | 72.97 | 73.41 | 73.11 | 73.39 | 73.32 | 73.43 | 73.56 | 63.16 | 73.51 | 75.69 | 75.64 | 73.37 | 73.70 | 73.27 | 73.22 | 73.94 | 73.58 |
| 11 | 98.37 | 97.81 | 98.76 | 98.78 | 98.52 | 98.73 | 98.46 | 98.46 | 98.56 | 98.34 | 98.40 | 98.60 | 98.48 | 98.41 | 98.48 | 95.94 | 98.39 | 98.68 | 98.59 | 98.52 | 98.62 | 98.56 | 98.27 | 98.62 | 98.80 |
| 12 | 55.74 | 52.39 | 58.60 | 58.93 | 57.30 | 58.67 | 57.90 | 57.91 | 58.81 | 57.72 | 57.95 | 58.50 | 58.29 | 58.38 | 57.80 | 52.73 | 52.81 | 58.23 | 58.56 | 58.45 | 58.28 | 58.01 | 55.58 | 58.70 | 58.56 |
| 17 | 78.31 | 72.80 | 81.21 | 81.33 | 80.14 | 81.55 | 81.53 | 81.18 | 81.17 | 79.74 | 81.27 | 81.31 | 81.00 | 81.53 | 81.50 | 81.39 | 81.12 | 81.14 | 81.08 | 81.42 | 81.36 | 80.80 | 75.99 | 81.45 | 81.37 |
| 18 | 83.85 | 81.65 | 86.97 | 87.59 | 86.92 | 86.31 | 86.56 | 86.56 | 86.54 | 86.31 | 86.52 | 87.09 | 86.74 | 86.74 | 86.82 | 83.13 | 86.24 | 87.43 | 85.42 | 87.43 | 87.35 | 87.06 | 85.79 | 86.75 | 87.94 |
| 19 | 66.79 | 70.82 | 68.58 | 67.61 | 68.91 | 68.53 | 69.75 | 70.35 | 65.42 | 65.77 | 70.75 | 70.15 | 70.02 | 70.12 | 69.58 | 69.28 | 66.94 | 70.50 | 68.11 | 69.30 | 70.50 | 69.70 | 70.42 | 68.03 | 70.27 |
| 25 | 85.22 | 84.13 | 90.00 | 90.82 | 87.65 | 90.70 | 88.47 | 88.51 | 87.30 | 87.87 | 88.30 | 89.23 | 88.47 | 88.58 | 88.43 | 85.01 | 87.43 | 91.35 | 90.53 | 88.78 | 89.36 | 89.15 | 87.31 | 89.53 | 89.90 |
| 26 | 67.37 | 68.78 | 73.43 | 73.40 | 73.24 | 73.37 | 73.35 | 73.07 | 73.20 | 71.47 | 73.15 | 73.22 | 73.26 | 73.21 | 73.28 | 67.99 | 66.77 | 73.08 | 73.20 | 73.25 | 73.29 | 73.17 | 72.54 | 73.15 | 73.24 |
| 27 | 83.25 | 79.21 | 83.25 | 83.26 | 83.25 | 83.32 | 83.34 | 83.28 | 83.31 | 83.19 | 83.21 | 83.33 | 83.30 | 83.27 | 83.32 | 83.20 | 83.26 | 83.22 | 83.31 | 83.33 | 83.33 | 83.36 | 83.15 | 83.31 | 83.32 |
| 28 | 87.92 | 89.17 | 91.35 | 90.57 | 87.97 | 91.57 | 92.67 | 94.07 | 93.28 | 83.24 | 93.19 | 91.51 | 93.72 | 93.71 | 92.47 | 92.94 | 91.60 | 92.72 | 94.94 | 92.08 | 93.33 | 90.03 | 94.44 | 90.97 | 93.60 |
| 29 | 89.58 | 87.29 | 91.60 | 91.33 | 88.51 | 91.33 | 92.88 | 93.72 | 93.06 | 85.99 | 93.82 | 90.82 | 93.43 | 93.50 | 93.15 | 92.82 | 91.39 | 94.36 | 94.35 | 91.93 | 94.29 | 90.31 | 94.64 | 90.83 | 94.15 |
| 30 | 86.46 | 87.29 | 90.11 | 89.86 | 88.78 | 89.99 | 93.22 | 92.61 | 84.94 | 92.15 | 90.06 | 91.53 | 92.57 | 92.22 | 91.74 | 90.68 | 93.93 | 93.42 | 92.17 | 92.62 | 89.21 | 93.51 | 90.67 | 92.64 | |
| 31 | 86.46 | 85.62 | 90.14 | 90.21 | 88.92 | 90.39 | 91.85 | 92.15 | 91.81 | 85.58 | 92.15 | 89.00 | 91.31 | 92.17 | 91.94 | 91.29 | 90.11 | 92.96 | 93.22 | 89.35 | 92.78 | 88.79 | 92.86 | 89.24 | 92.86 |
| 32 | 86.04 | 87.50 | 88.19 | 87.51 | 87.15 | 88.10 | 91.29 | 92.64 | 91.54 | 82.06 | 91.72 | 88.96 | 91.25 | 92.85 | 91.60 | 91.61 | 88.99 | 89.49 | 93.81 | 89.47 | 92.22 | 86.68 | 92.75 | 86.68 | 92.64 |
| 34 | 77.86 | 63.11 | 78.01 | 79.72 | 58.54 | 77.12 | 77.56 | 76.98 | 71.94 | 70.34 | 77.08 | 77.70 | 77.12 | 77.71 | 76.98 | 76.77 | 64.23 | 82.01 | 79.86 | 78.91 | 78.50 | 78.41 | 66.76 | 79.06 | 75.47 |
| 35 | 86.61 | 88.90 | 88.41 | 89.05 | 87.71 | 89.29 | 88.74 | 89.06 | 85.80 | 85.30 | 88.07 | 89.27 | 89.08 | 89.17 | 88.26 | 88.24 | 86.92 | 89.51 | 89.14 | 88.75 | 89.40 | 89.14 | 89.32 | 89.43 | 89.51 |
| 37 | 62.34 | 72.27 | 72.67 | 74.14 | 73.68 | 72.01 | 74.81 | 74.23 | 57.72 | 60.87 | 75.61 | 73.19 | 71.60 | 76.71 | 73.30 | 71.77 | 66.44 | 75.04 | 74.69 | 74.46 | 73.74 | 73.80 | 72.61 | 75.01 | 75.82 |
| 39 | 62.55 | 63.64 | 67.45 | 66.72 | 66.60 | 67.54 | 66.32 | 66.46 | 66.34 | 66.28 | 65.33 | 67.28 | 66.50 | 66.27 | 65.84 | 62.92 | 64.76 | 67.00 | 67.97 | 67.41 | 66.80 | 66.94 | 65.02 | 66.88 | 67.28 |
| 40 | 79.81 | 72.50 | 84.77 | 85.00 | 85.13 | 84.93 | 85.03 | 84.81 | 84.38 | 81.57 | 85.08 | 84.77 | 84.86 | 84.97 | 84.65 | 77.03 | 84.60 | 83.82 | 84.59 | 85.07 | 84.35 | 85.06 | 85.01 | 84.53 | 84.68 |
| 41 | 71.90 | 72.71 | 73.96 | 74.27 | 73.65 | 74.72 | 73.25 | 73.40 | 74.51 | 73.10 | 73.43 | 74.10 | 73.93 | 73.74 | 73.42 | 73.53 | 72.75 | 73.63 | 74.28 | 73.99 | 74.21 | 74.11 | 73.82 | 73.85 | 74.04 |
| 42 | 62.13 | 73.91 | 67.14 | 68.16 | 67.50 | 68.12 | 73.84 | 74.80 | 55.77 | 67.87 | 72.77 | 73.46 | 70.90 | 73.70 | 71.93 | 73.86 | 70.10 | 74.28 | 70.57 | 69.45 | 74.22 | 68.07 | 75.00 | 64.80 | 75.26 |
| 43 | 57.65 | 70.34 | 63.81 | 66.30 | 58.49 | 66.42 | 71.21 | 71.21 | 59.00 | 71.06 | 71.12 | 69.01 | 70.89 | 71.08 | 71.14 | 70.70 | 70.10 | 71.16 | 71.15 | 70.08 | 71.42 | 70.52 | 69.89 | 69.78 | 71.13 |
| 44 | 59.40 | 75.20 | 66.27 | 63.98 | 65.54 | 66.18 | 69.01 | 74.64 | 65.23 | 51.35 | 53.70 | 51.79 | 65.67 | 74.80 | 72.62 | 74.24 | 67.54 | 75.20 | 75.44 | 65.00 | 75.19 | 52.97 | 75.59 | 62.05 | 75.55 |
| 45 | 59.13 | 76.64 | 64.87 | 65.58 | 64.16 | 64.60 | 73.04 | 76.02 | 55.06 | 56.71 | 67.16 | 67.03 | 68.83 | 74.50 | 74.02 | 73.13 | 69.41 | 76.55 | 75.26 | 67.74 | 77.09 | 64.21 | 75.89 | 63.05 | 76.17 |
| 46 | 54.80 | 71.42 | 64.07 | 64.95 | 58.18 | 64.40 | 71.73 | 71.78 | 54.36 | 58.69 | 70.58 | 69.13 | 70.06 | 71.32 | 71.38 | 71.32 | 65.47 | 72.02 | 72.09 | 67.77 | 72.08 | 69.28 | 67.25 | 70.61 | 71.36 |
| 47 | 52.54 | 70.41 | 60.23 | 61.58 | 56.53 | 62.28 | 69.85 | 70.92 | 52.25 | 64.13 | 65.62 | 61.84 | 68.84 | 69.09 | 68.99 | 66.99 | 63.36 | 70.80 | 71.09 | 64.53 | 70.56 | 61.46 | 58.13 | 65.59 | 66.28 |
| 48 | 53.13 | 69.13 | 60.80 | 61.65 | 56.57 | 62.25 | 68.31 | 69.32 | 53.47 | 67.62 | 64.94 | 67.17 | 65.59 | 69.76 | 68.82 | 69.65 | 62.14 | 71.09 | 70.98 | 61.62 | 70.84 | 56.90 | 54.51 | 67.46 | 59.37 |
| 49 | 52.04 | 71.39 | 60.25 | 58.89 | 57.13 | 59.13 | 72.04 | 70.05 | 70.63 | 56.71 | 70.14 | 66.65 | 67.67 | 71.52 | 59.26 | 71.50 | 65.45 | 71.24 | 70.41 | 61.36 | 73.32 | 62.65 | 73.08 | 63.65 | 72.84 |
| 50 | 54.61 | 70.99 | 61.21 | 62.59 | 57.18 | 62.72 | 70.61 | 70.96 | 53.38 | 62.43 | 68.71 | 68.80 | 69.71 | 70.45 | 69.90 | 70.34 | 65.86 | 64.32 | 71.18 | 63.79 | 71.31 | 66.44 | 62.90 | 70.26 | 69.15 |
| 52 | 80.37 | 77.62 | 85.71 | 85.97 | 80.72 | 85.76 | 86.74 | 87.11 | 81.42 | 83.01 | 85.79 | 87.11 | 86.48 | 86.39 | 86.40 | 86.36 | 86.20 | 87.34 | 87.19 | 86.60 | 87.20 | 85.30 | 83.86 | 87.15 | 87.08 |
| 53 | 74.00 | 76.67 | 75.38 | 78.11 | 77.93 | 79.64 | 76.29 | 77.56 | 78.62 | 75.11 | 76.18 | 77.02 | 76.24 | 76.93 | 77.04 | 74.58 | 75.38 | 78.04 | 78.40 | 77.24 | 77.69 | 74.04 | 79.93 | 78.98 | 78.56 |
| 55 | 55.94 | 48.77 | 60.35 | 68.08 | 58.96 | 61.02 | 59.40 | 60.23 | 56.96 | 49.38 | 55.85 | 49.31 | 54.12 | 58.90 | 63.65 | 64.33 | 57.75 | 63.77 | 59.90 | 55.40 | 65.31 | 50.98 | 58.94 | 48.85 | 65.69 |
| 56 | 66.19 | 50.19 | 67.12 | 67.96 | 65.44 | 68.23 | 65.12 | 65.71 | 53.29 | 49.54 | 63.88 | 53.25 | 59.62 | 64.34 | 64.92 | 64.83 | 67.79 | 65.71 | 63.19 | 67.33 | 52.06 | 65.56 | 58.00 | 68.35 | |
| 57 | 57.26 | 58.88 | 59.87 | 59.26 | 59.59 | 58.82 | 60.09 | 59.61 | 59.17 | 58.74 | 59.00 | 59.86 | 59.51 | 58.95 | 59.57 | 58.09 | 58.84 | 59.14 | 59.25 | 59.48 | 59.81 | 59.41 | 59.00 | 59.77 | 59.85 |
| 58 | 65.67 | 45.58 | 68.94 | 69.28 | 62.14 | 68.33 | 63.42 | 61.76 | 52.30 | 49.26 | 61.34 | 65.65 | 63.70 | 62.50 | 63.83 | 57.67 | 54.99 | 73.92 | 69.51 | 66.79 | 69.43 | 68.44 | 55.93 | 69.80 | 66.06 |
| 60 | 79.10 | 78.05 | 79.37 | 79.51 | 77.96 | 79.34 | 79.53 | 79.29 | 79.10 | 78.32 | 79.38 | 79.80 | 79.60 | 79.72 | 79.22 | 79.68 | 77.32 | 79.84 | 79.58 | 79.80 | 79.56 | 79.40 | 77.35 | 79.82 | 79.74 |
| 61 | 93.70 | 85.77 | 90.78 | 93.47 | 92.18 | 91.29 | 89.10 | 88.32 | 91.62 | 83.78 | 88.01 | 88.94 | 87.11 | 88.21 | 88.15 | 87.65 | 90.50 | 87.98 | 91.34 | 89.08 | 91.15 | 85.46 | 90.78 | 87.42 | 92.30 |
| 63 | 84.16 | 80.64 | 89.71 | 89.21 | 87.30 | 89.45 | 87.37 | 87.21 | 84.43 | 86.20 | 86.37 | 88.50 | 87.19 | 87.46 | 87.18 | 85.30 | 78.41 | 89.47 | 89.70 | 88.53 | 88.88 | 88.25 | 83.37 | 89.10 | 87.73 |
| 64 | 89.50 | 89.87 | 92.32 | 92.07 | 89.05 | 84.27 | 93.01 | 94.63 | 72.64 | 93.38 | 91.54 | 93.35 | 92.83 | 93.43 | 93.97 | 91.21 | 36.89 | 96.50 | 96.85 | 93.29 | 95.45 | 93.39 | - | 94.90 | 95.45 |
| 65 | 75.75 | 75.75 | 75.75 | 75.75 | 75.75 | 75.75 | 75.75 | 75.73 | 75.74 | 75.76 | 75.70 | 75.74 | 75.70 | 75.74 | 75.66 | 75.69 | 75.73 | 75.71 | 75.73 | 75.68 | 75.71 | 75.71 | 75.70 | 75.72 | 75.76 |
| 66 | 92.59 | 92.55 | 93.80 | 94.72 | 94.03 | 94.18 | 94.41 | 94.28 | 94.59 | 93.38 | 94.45 | 94.26 | 94.34 | 94.52 | 94.31 | 94.02 | 94.08 | 94.45 | 94.18 | 94.27 | 94.29 | 94.16 | 93.70 | 94.35 | 94.24 |
| 67 | 88.70 | 88.08 | 88.27 | 88.41 | 88.41 | 88.46 | 88.31 | 88.02 | 87.94 | 88.25 | 88.60 | 88.39 | 87.85 | 88.37 | 88.17 | 88.25 | 87.67 | 88.62 | 88.43 | 88.58 | 88.17 | 88.41 | 87.85 | 88.35 | |
| 68 | 98.60 | 92.83 | 98.03 | 98.30 | 96.05 | 98.21 | 98.98 | 99.35 | 94.81 | 97.25 | 98.76 | 98.69 | 98.79 | 98.79 | 98.89 | 96.58 | 98.10 | 99.59 | 99.56 | 98.70 | 99.40 | 98.31 | 97.27 | 98.99 | 98.56 |

```
70  71.70 74.85 80.77 80.79 80.01 80.21 77.06 77.17 80.40 76.76 76.35 80.28 76.50 77.10 76.72 73.55 74.63 79.72 79.58 78.61 79.25 80.25 77.58 80.33 81.34
71  69.79 73.39 81.38 81.03 80.48 80.34 75.52 75.02 77.41 76.84 75.00 80.38 76.51 75.65 76.40 74.75 71.68 79.41 79.63 79.81 79.43 80.70 76.82 80.27 81.14
77  89.51 99.08 99.12 98.82 96.60 99.15 97.79 98.60 99.43 98.44 98.40 98.87 98.36 99.01 98.56 95.31 95.67 98.99 98.99 99.11 99.45 99.00 98.53 99.65 99.76
78  84.85 87.88 89.37 88.71 87.35 91.56 88.06 88.71 84.91 84.06 86.87 89.68 87.78 89.09 88.85 89.09 86.97 88.48 87.82 88.81 87.96 88.95  -    90.14 89.70
80  84.65 79.34 87.63 87.87 86.90 87.88 87.24 87.61 87.01 86.98 86.91 87.59 87.03 87.18 86.88 83.09 86.70 88.13 85.96 87.05 87.86 87.58 86.69 87.83 88.24
81  87.95 87.96 88.17 88.05 87.22 88.05 87.71 87.65 86.15 84.94 88.12 86.88 88.26 86.43 88.85 87.72 86.95 87.95 90.19 88.18 88.04 88.04 88.88 88.50 89.08
84  67.25 82.92 90.80 93.57 87.02 91.08 91.57 92.65 89.53 88.07 91.75 93.67 93.73 95.50 95.42 57.23 84.50 96.23 98.12 96.97 96.27 95.27 96.77 96.88 97.17
88  82.89 83.19 84.23 84.27 85.07 83.64 83.03 82.56 84.25 83.77 83.30 83.63 83.83 83.00 83.04 83.08 83.20 83.44 83.26 83.49 83.85 83.39 83.38 83.41 83.70
92  72.39 69.23 71.17 72.91 72.04 70.90 72.11 73.38 72.19 66.47 68.86 72.14 72.09 73.06 71.59 67.86 70.10 70.95 72.89 69.53 71.59 70.75 75.52 72.59 75.45
93  96.61 93.98 97.22 96.79 96.39 97.19 97.27 97.40 94.86 94.95 97.33 96.76 96.94 96.96 97.09 97.30 96.20 97.74 97.67 97.33 97.25 96.98 94.95 97.47 96.47
96  71.43 74.68 76.45 77.75 77.27 78.14 76.06 74.94 77.06 73.03 72.51 75.58 76.62 73.64 76.49 68.61 76.02 75.97 75.24 75.76 74.03 76.58 75.58 77.01 77.40
98  86.40 85.87 87.16 87.25 87.40 87.47 86.91 86.77 86.07 86.13 85.08 86.72 87.07 87.17 86.49 86.39 86.33 86.87 86.99 87.01 86.96 86.85 87.45 87.25 87.27
99  89.10 84.42 85.06 87.30 87.42 84.08 86.67 86.67 69.76 79.87 86.86 86.95 85.12 87.08 87.71 86.26 85.69 86.03 84.55 87.27 85.88 86.89 87.52 84.93 87.84
100 83.31 82.21 85.28 86.89 76.55 85.18 85.85 85.73 84.74 83.74 84.33 85.66 85.50 85.91 85.40 85.02 81.78 86.09  -    86.27 86.96 85.05 83.12 86.78 86.35
102 72.72 93.55 97.67 97.52 81.77 97.60 94.58 95.27 97.28 97.25 97.10 97.03 95.02 96.83 96.62 91.96 76.52 97.24 97.38 97.22 97.21 97.40 96.53 97.21 97.16
103 62.18 64.22 66.83 67.00 66.30 66.75 65.93 67.43 66.68 66.38 65.39 68.48 66.60 67.64 67.10 64.21 64.80 67.98 67.08 67.17 67.50 67.98 65.68 67.74 68.49
106 85.46 86.10 95.50 95.17 94.22 95.80 87.94 89.09 91.29 89.32 87.36 95.19 92.58 91.96 91.29 85.74 90.85 94.09 93.73 93.58 94.16 93.20 93.30 94.78 95.32
108 98.95 98.66 98.95 98.95 98.95 98.95 98.92 98.82 98.95 98.81 98.93 98.93 98.95 98.92 98.92 98.82 98.91 98.88 98.79 98.90 98.89 98.93 98.95 98.94 98.95
109 63.87 60.98 61.44 62.50 62.93 62.86 63.01 62.80 65.05 60.66 62.58 63.10 62.13 62.58 61.15 62.35 61.32 62.63 62.80 62.33 61.64 62.23 61.54 61.68 62.69
110 67.23 67.96 65.88 65.10 65.27 65.80 68.74 67.93 70.56 58.15 68.60 65.35 66.25 68.29 68.49 67.54 69.05 65.97 70.03 66.30 69.80 60.06 74.45 62.16 74.51
111 96.58 96.65 98.83 98.79 98.36 98.69 98.85 98.42 94.61 97.11 98.46 98.49 98.94 98.51 98.35 97.61 98.69 98.94 99.63 98.80 98.57 99.14 98.22 99.12
114 59.57 61.87 61.52 61.84 61.90 60.97 63.15 63.57 59.62 61.18 63.29 63.01 62.73 62.62 62.49 60.68 62.11 62.71 63.04 62.54 63.25 62.93 61.98 62.95 63.51
116 82.89 83.61 85.53 85.66 84.76 85.12 84.34 84.45 82.75 83.02 84.07 84.80 83.39 84.05 83.08 84.46 82.86 84.83 84.97 84.73 84.35 84.72 84.42 85.73 85.61
117 79.74 83.07 83.62 84.51 84.54 83.82 83.48 82.92 81.34 80.25 82.63 83.69 83.64 83.25 83.39 81.80 82.38 83.57 83.21 83.96 83.37 83.87 84.05 84.05 84.25
118 80.94 84.34 85.03 85.00 83.86 84.49 82.98 82.56 81.55 82.67 82.21 84.83 84.45 84.15 83.69 84.00 84.05 84.57
119 84.07 82.72 86.93 87.31 86.23 87.41 85.19 85.16 86.06 85.25 85.35 85.88 85.61 85.69 85.72 84.65 85.68 86.90 87.28 86.39 86.10 86.02 84.44 87.19 86.46
121 63.50 60.50 61.67 62.88 63.17 61.27 57.20 58.05 58.65 55.22 59.02 60.32 60.05 60.97 61.43 63.63 59.88 62.42 60.08 62.17 60.93 59.13 58.40 62.37 60.80
122 96.16 95.81 98.03 97.78 96.83 97.60 96.49 96.50 95.87 96.58 96.44 97.09 97.26 96.30 96.94 96.20 96.10 97.04 97.13 97.16 97.13 97.46 97.29 97.28 97.65
123 97.09 96.77 98.22 97.99 97.73 98.53 96.67 97.02 96.69 96.59 97.01 97.73 97.43 97.03 97.62 96.62 96.27 97.93 97.79 97.81 97.67 97.60 96.99 98.56 98.13
124 98.82 99.41 98.78 98.58 98.66 99.09 98.70 97.71 97.99 90.53 98.62 97.24 98.42 98.93 98.74 98.38 98.19 98.86 99.41 98.78 99.33 97.95 99.37 99.13 99.37
128 83.32 32.13 87.72 88.96 76.57 88.51 78.10 77.14 60.77 63.88 70.36 80.80 76.12 77.91 77.27 61.29 71.83 90.38 91.67 91.32 83.27 77.76 63.04 92.65 65.56
130 54.40 60.53 65.72 66.49 61.07 64.33 58.39 58.47 51.04 45.23 57.85 62.77 59.17 57.65 52.81 58.01 46.51 57.91 50.77 55.87 62.71 57.64 50.15 61.11 65.39
131 35.00 31.21 35.45 35.79 35.48 39.24 34.00 35.58 34.73 23.36 33.88 35.55 32.36 36.36 32.67 34.15 24.27 35.64 34.45 32.27 33.61 30.09 34.21 34.58 36.76
133 89.42 90.19 90.60 90.82 89.98 90.63 90.52 90.60 90.35 89.99 90.45 90.95 90.69 90.50 90.70 89.99 90.25 90.87 90.74 90.71 90.95 90.92 89.26 90.89 90.97
136 55.64 53.09 48.12 51.85 47.32 56.41 53.33 52.87 53.31 51.71 55.13 54.76 55.22 54.35 52.12 54.55 52.75 54.04 52.75 53.21 53.24 53.62 55.64 51.81 55.64
137 92.54 95.52 93.98 93.88 94.60 94.20 92.54 93.13 91.64 92.31 92.89 93.76 92.76 93.58 93.86 90.17 94.15 94.73 92.74 94.10 93.68 93.33 93.33 94.13 94.45
139 88.73 89.50 98.09 99.23 95.92 98.90 98.15 98.48 75.57 89.85 98.94 98.21 98.65 97.98 98.50 99.50 67.61 99.34 99.46 98.75 99.54 97.32 97.51 98.02 99.17
141 89.30 86.25 94.88 95.75 94.39 94.65 93.25 93.84 93.32 89.97 93.05 96.34 95.51 94.79 89.29 92.15 95.75 95.69 95.15 96.30 95.13 93.53 94.36
142 49.15 54.58 56.14 57.02 55.82 57.51 55.89 56.29 55.03 46.87 54.46 56.72 56.61 57.29 56.47 47.71 47.25 56.27 55.66 54.78 56.88 57.27 57.31 55.21 57.47
146 96.48 96.74 96.98 96.89 96.99 97.01 96.79 96.64 97.09 96.74 96.85 96.65 96.87 96.84 96.75 96.87 96.71 96.83 97.89 96.63 96.81 96.88 97.00 96.71 96.92
147 75.88 68.13 78.45 76.37 74.69 75.08 75.93 76.60 69.81 69.07 74.69 76.64 75.92 73.59 72.64 75.01 65.59 86.64 85.17 77.09 79.04 71.45 69.36 78.79 73.73
149 77.14 65.95 85.17 85.17 73.82 83.97 82.40 83.40 83.81 81.51 84.90 85.55 85.48 86.18 86.02  -   87.33 86.67 85.63 82.77 86.49 84.89 66.04 87.33 82.58
150 49.83 51.30 58.80 60.16 56.63 60.52 57.42 56.66 61.06 50.98 55.25 47.50 58.24 59.53 56.81 58.00 52.09 59.39 57.24 59.30 58.31 56.81 59.16 61.31 61.33
153 67.87 70.98 78.17 77.96 77.95 77.82 75.80 75.40 62.25 74.69 75.54 75.88 75.61 75.56 75.28 74.67 74.43 77.55 72.98 77.23 77.82 75.78 76.93 77.43 77.65
154 72.50 75.50 76.83 77.67 76.70 74.50 73.30 75.77 78.07 68.37 75.33 79.13 74.33 78.97 77.40 75.83  -   75.40 78.50 74.23 74.43 78.43 74.80 78.13 78.63
155 93.43 90.23 96.07 96.31 93.36 96.06 96.00 95.99 95.57 95.82 95.53 95.05 96.34 96.17 96.06 96.08 95.95 91.32 96.44 95.71
156 60.81 63.01 64.20 64.62 62.18 64.17 62.97 63.14 63.91 62.53 63.13 63.87 64.06 64.10 64.22 62.47 63.76 63.86 63.96 63.95 63.93 63.79 61.45 63.84 63.60
160 94.67 94.31 95.02 94.82 94.89 94.85 94.49 94.73 94.44 94.35 94.69 94.71 94.73 94.67 94.42 90.68 94.60 94.82 94.93 94.69 94.52 94.60 94.72 94.95 94.98
162 98.41 98.41 98.76 98.66 98.41 98.70 98.52 98.41 98.41 98.37 98.35 98.76 98.48 98.39 98.50 98.41 98.40 98.76 98.68 98.42 98.87 98.58 98.41 98.71 99.00
163 89.66 96.08 96.30 96.52 94.97 96.99 92.24 96.33 90.79 91.72 93.39 93.50 95.57 95.02 95.97 95.61  -   94.85 95.65 93.73 96.54 96.51  -   96.76
164 40.58 39.26 38.59 38.44 39.35 39.49 38.51 37.15 40.27 37.97 35.90 39.86 39.06 36.92 39.40 38.94 36.82 39.26 39.89 37.79 38.13 39.45 40.32 39.58 40.32
165 91.48 91.14 92.10 92.18 91.29 92.29 92.46 92.54 91.07 91.67 92.24 92.53 92.47 92.49 92.38 92.18 35.09 92.74 92.58 92.42 92.64 92.41 92.19 92.45 92.37
166 92.22 62.48 94.49 95.68 88.41 94.60 95.69 93.61 81.95 63.12 94.56 95.35 95.26 95.64 95.73 91.70 81.56 98.19 99.11 95.42 98.95 93.67 92.84 96.90 95.49
167 84.27 76.01 92.71 91.69 85.26 92.31 85.37 85.52 83.18 81.92 84.95 87.51 86.05 85.55 85.15 79.49 82.84 96.39 95.62 86.82 87.70 87.44 79.45 88.66 84.99
169 84.63 73.74 90.06 94.69 93.18 96.08 93.24 93.74 75.43 92.10 94.29 97.13 95.24 95.19 96.75 86.70 80.80 98.03 98.28 97.42 95.92 96.83 96.78 97.49 97.75
170 59.46 64.19 67.66 67.70 67.66 65.72 65.81 68.83 70.90 48.29 61.85 67.12 67.43 66.71 66.13 61.44  -   68.24 70.77 68.87 67.97 69.73 69.77 68.06 72.39
171 61.24 49.56 72.64 71.88 64.62 72.85 62.77 63.71 60.82 54.80 59.63 71.59 61.15 62.87 63.33 54.58 43.34 96.98 88.55 80.07 76.66 59.58 55.06 92.71 75.26
172 58.02 55.58 63.25 61.52 60.57 63.93 57.08 57.54 56.11 56.45 57.45 59.45 58.31 57.41 57.64 56.73 55.79 65.95 90.88 63.30 59.63 60.28 57.84 59.87 63.16
174 59.89 48.45 61.23 61.11 61.66 60.11 57.63 58.42 43.57 50.95 57.10 57.51 57.03 57.12 57.76 51.26 55.80 58.91 59.95 56.48 57.49 55.24 57.75 56.08 58.14
177 99.46 98.96 99.42 99.49 99.40 99.47 99.41 99.50 98.80 98.39 99.35 99.20 99.45 99.40 99.34 99.26 48.31 99.48 99.49 99.57 99.66 99.24  -   99.63 99.48
179 84.02 90.87 92.91 91.05 91.45 91.99 88.31 88.92 91.84 89.04 88.77 92.88 94.40 89.92 94.52 84.35 88.31 93.55 91.29 90.96 90.68 91.29 90.81 93.00 93.06
182 71.45 71.25 72.33 72.30 71.98 72.21 71.81 71.75 72.05 71.21 71.94 72.17 71.95 70.87 71.63 71.87 71.40 72.17 72.13 72.40 72.17 72.45 71.75 72.21 72.29
183 55.56 78.68 57.78 51.66 54.98 60.27 65.49 77.64 49.19 49.82 61.26 50.81 54.87 59.53 62.69 79.44 82.84 78.11 53.44 79.64 54.54 51.63 52.67 63.43
184 86.32 81.46 88.26 88.47 87.88 88.08 87.17 86.97 87.23 85.03 86.86 87.47 87.21 86.85 86.62 86.26 86.32 87.02 87.04 86.83 87.44 87.22 87.11 87.41 87.75
191 97.99 97.98 97.89 97.94 98.03 97.81 97.83 97.86 97.54 97.83 97.80 97.89 97.83 97.86 97.82 97.85 97.91 97.85 98.00 97.94 97.85 97.93 97.80 97.97 97.95
192 85.03 99.95 100.0 100.0 100.0 100.0 95.06 96.17 84.49 92.79 93.67 100.0 93.65 97.80 97.23 92.27 86.12 99.98 100.0 99.93 99.84 100.0 99.86 100.0 100.0
193 69.53 67.09 77.10 76.22 74.42 76.85 76.29 76.65 72.25 67.77 75.52 75.70 74.90 75.70 76.62 73.99 76.83 75.90 75.57 76.65 74.60 64.39 77.18 75.57
194 93.10 79.54 92.98 93.08 93.11 93.09 80.12 76.80 79.76 73.56 79.84 79.73 79.94 80.02 79.87 74.12 76.18 93.09 92.92 80.45 93.10 79.94 81.59 80.26 80.18
196 78.39 77.05 80.29 79.72 81.42 80.89 75.02 76.64 78.22 76.68 75.31 78.92 77.98 76.87 77.54  -   75.63 80.04 78.75 78.05 76.91 78.88  -   79.65 79.30
197 81.67 81.29 81.44 80.84 81.11 80.89 81.25 80.95 81.31 80.94 80.78 81.41 81.35 81.09 81.00 81.31 81.24 81.65 80.90 81.28 81.04 80.86 81.32 81.53 81.27
198 88.14 68.01 86.10 86.47 82.62 86.59 95.27 86.96 86.51 87.84 92.11 97.61 95.52 96.74 90.57 71.44 64.80 98.98 99.50 99.07 99.58 97.34 82.10 94.41 96.99
199 86.02 85.07 86.27 86.24 86.98 87.44 85.81 85.94 85.02 84.11 85.40 85.34 86.26 85.53 84.88 86.07 85.96 84.74 84.53 85.26 86.10 84.71 86.62 86.03 86.73
200 83.72 85.81 88.36 87.74 88.27 88.70 84.96 85.16 83.28 84.69 84.89 84.87 84.58 85.07 85.07 84.96 84.23 87.60 88.14 87.87 87.73 84.82 85.99 87.41 86.03
203 93.59 97.94 99.84 99.84 99.27 99.84 99.40 99.52 99.77 96.02 99.43 99.69 99.68 99.67 99.54 99.43 99.38 99.78 99.76 99.60 99.69 99.51 99.24 99.44 99.80
205 93.82 100.0 100.0 100.0 100.0 100.0 99.99 100.0 100.0 99.93 100.0 98.02 100.0 100.0 99.44 94.17  -   100.0 100.0 100.0 100.0 94.39 100.0 100.0
206 72.34 73.44 73.01 73.49 73.11 72.41 73.78 73.74 73.49 70.73 72.78 73.31 73.54 72.59 73.26 73.62 73.39 73.83 73.18 72.78 73.41 73.70 73.55 73.41 73.47
207 72.97 72.03 74.21 74.31 74.50 74.52 73.97 74.16 73.93 72.33 73.50 73.61 73.89 73.48 73.60 73.58 73.97 73.66 73.88 73.78 73.33 73.45 73.41 73.94 73.99
209 67.83 60.35 86.32 86.43 74.12 86.38 61.52 59.38 76.45 55.30 58.56 66.91 57.31 59.17 61.41 59.52 50.91 72.40 83.32 62.48 84.67 58.27  -   64.13 82.60
210 98.66 98.19 98.67 98.83 98.52 98.76 98.67 98.71 98.21 98.31 98.78 98.71 98.70 98.65 98.70 98.17 98.43 98.72 98.56 98.71 98.81 98.64 98.65 98.69 98.75
211 82.76 62.56 82.13 81.28 82.04 83.32 72.41 76.29 73.92 66.40 72.91 71.23 68.01 66.77 71.66 66.83 81.05 83.25 82.99 81.25 82.46 73.60 77.50 80.43 78.89
213 96.25 97.75 96.60 96.80 96.55 96.88 96.15 97.25 93.83 91.23 96.02 96.42 95.93 95.77 96.32 96.63 95.80 96.93 97.22 96.37 97.10 96.53  -   96.57 96.55
```

```
214 86.50 83.50 85.72 87.18 86.02 86.23 84.45 85.97 83.30 77.42 84.07 85.23 83.90 84.67 83.88 81.88 84.25 86.53 86.48 85.23 88.08 85.55 83.63 86.72 87.73
215 96.00 94.25 94.00 95.95 94.72 95.02 95.63 96.50 93.77 84.10 95.42 95.55 95.27 95.35 94.83 94.72 94.48 95.57 96.53 95.75 96.42 94.95 94.62 96.07 96.58
216 76.00 74.92 74.88 74.28 74.90 75.33 76.43 76.42 69.00 74.53 76.92 76.68 76.37 75.92 76.17 76.25 73.92 76.33 75.53 76.15 75.43 74.45 76.10 76.00 76.67
217 92.25 93.75 96.13 95.20 95.90 95.92 95.47 95.65 94.55 86.85 94.52 94.90 95.85 96.00 95.60  -   96.00 96.37 96.50 95.08 95.88 95.00  -   96.00 94.08
218 80.50 82.38 78.07 78.55 76.03 79.45 84.77 83.62 67.93 72.63 83.47 83.33 83.17 82.98 82.50 81.72 81.03 87.27 79.02 82.18 83.03 79.77 81.58 83.90 81.97
219 100.0 98.61 96.30 99.07 97.96 98.61 99.23 99.94 98.80 85.31 99.14 98.02 98.86 99.48 99.32 99.35 99.78 99.88  -   98.12 100.0 97.96 99.60 98.33 100.0
220 58.63 56.80 62.03 61.89 61.12 61.92 63.00 62.93 61.16 61.42 62.20 62.76 62.18 62.90 62.22 59.67 46.39 63.28 63.08 62.77 63.79 62.59 60.60 62.39 63.70
221 96.68 83.69 98.05 98.03 97.01 97.95 98.32 98.04 96.13 97.19 98.07 98.17 98.21 98.26 98.25 88.56 83.55 98.93 98.47 98.33 98.47 98.14 93.40 98.53 98.27
222 91.12 77.85 94.59 94.66 94.56 94.58 91.94 91.35 92.73 92.61 91.48 92.76 90.90 92.50 82.88 85.09 91.53 94.26 94.28 93.86 93.24 92.79 91.19 93.47 93.10
224 96.56 95.81 96.80 96.95 96.34 96.97 96.87 96.94 96.19 96.55 96.93 96.91 96.92 96.81 96.54 97.00 96.96 97.02 96.90 96.98 96.93 96.40 96.31 97.03 96.86
225 78.92 98.68 100.0 100.0 100.0 100.0 94.53 96.45 99.79 98.90 94.18 99.65 99.37 98.45 98.30 91.93 90.72 99.40 99.89 99.88 99.53 97.73 99.50 100.0 100.0
226 72.68 74.25 75.12 75.08 72.27 74.75 74.50 74.28 73.09 73.36 74.58 75.00 74.55 74.81 74.83 73.62 71.85 74.81 74.75 74.71 74.79 74.07 73.12 74.78 74.98
230 84.59 87.79 90.51 90.63 86.33 90.19 88.03 88.71 88.75 89.08 88.64 90.19 89.76 90.32 89.69 88.68 88.29 89.65 90.53 90.21 90.11 89.98 89.69 89.49 89.89
231 97.95 95.82 98.24 98.24 98.01 98.31 98.25 98.64 95.71 94.33 98.05 97.79 97.98 98.35 98.28 98.07 98.19 98.71 99.13 97.85 98.70 96.98 97.89 98.62 98.35
232 95.07 94.08 94.79 94.40 95.00 94.81 93.74 93.74 93.69 93.46 93.03 94.83 94.53 94.31 94.00 93.52 94.46 94.42 94.66 93.37 94.02 94.52 95.13 94.94 95.12
233 98.23 96.65 98.14 97.81 98.23 97.87 97.43 97.68 98.12 98.06 97.26 97.53 97.57 97.47 97.52 98.08  -   97.91 97.76 96.73 97.30  -   97.83 98.07 98.04
234 96.16 95.65 97.45 96.67 97.03 97.12 96.69 96.68 92.95 95.27 96.94 96.71 97.11 96.63 96.61 96.40 96.37 96.77 96.68 96.85 97.11 96.32 97.27 96.74 97.23
235 93.69 93.24 93.81 93.51 93.57 93.75 92.73 92.10 93.24 92.19 93.36 93.09 93.24 91.83 92.73 93.24 92.43 93.36 93.42 93.39 92.70 91.83 93.69 93.72 93.70
236 89.14 88.71 88.56 88.80 89.41 87.73 88.20 88.67 89.78 88.84 89.07 89.78 89.52 88.67 89.29 89.76 89.63 88.46 88.75 88.24 87.92 89.69 89.50 88.60 89.48
237 89.04 90.27 91.51 90.16 89.13 91.10 89.89 88.52 87.79 87.47 90.05 89.59 90.34 89.75 90.16 89.68 89.75 89.89 89.43 89.86 89.91 89.45 90.78 90.21 90.64
238 99.09 92.81 94.14 99.13 98.79 95.09 98.92 98.07 99.42 99.16 99.24 22.76 99.43 99.40 99.29 99.60 99.37 98.88 99.41 99.44
239 73.86 70.47 76.73 75.44 75.79 76.03 71.87 71.16 75.27 69.87 71.01 69.81 72.48 70.27 71.59 71.18 71.97 74.11 76.81 72.31 73.82 73.03  -   73.88 74.09
240 86.86 72.94 87.66 87.97 88.52 88.07 86.84 87.14 85.38 84.66 86.10 88.09 86.22 87.10 85.93 80.70 86.43 88.92 89.71 86.98 87.84 86.54 86.24 87.06 87.10
241 95.34 88.15 98.24 98.60 97.32 98.18 98.54 98.85 98.08 97.70 98.53 99.14 99.05 98.82 98.78 97.58 71.42 98.87 98.85 99.04 98.59 98.63 97.48 98.96 99.21
244 67.91 73.50 75.68 77.82 74.79 76.58 74.60 74.31 74.92 71.08 74.61 75.47 74.57 75.85 75.80 75.40 73.24 74.93 73.45 74.90 76.55 75.32 74.93 74.87 77.04
249 92.39 92.13 91.89 92.23 92.18 92.34 91.79 92.10 91.91 91.66 91.63 92.04 92.05 91.92 92.23 92.44 91.36 92.16 92.98 91.75 92.20 91.92 92.98 91.74 92.07
250 77.57 77.05 97.15 97.11 93.67 97.41 97.41 97.66 96.88 94.85 81.80 97.65 96.95 96.96 97.30 50.79 97.64 97.95 98.02 97.79 97.61 97.74 98.04 98.06 97.98
251 65.49 63.08 79.93 78.58 72.95 78.01 65.62 65.92 64.75 62.81 66.65 71.96 68.22 64.88 64.90 62.82 62.58 87.79 83.75 75.03 74.72 71.68 67.38 76.52 80.44
252 90.75 83.67 91.42 91.35 90.86 91.62 89.93 90.44 86.12 87.90 89.79 89.59 89.70 90.45 90.41 86.31 89.81 91.31 91.04 90.40 91.56 89.47 90.77 89.63 91.27
254 88.96 82.97 91.49 92.02 91.96 91.77 91.99 91.86 79.57 87.92 91.85 91.37 90.39 91.88 89.06 91.39 24.04 91.90 92.77 89.86 91.33 90.59 92.34 91.18 92.62
255 93.04 93.23 92.73 92.93 93.31 93.28 93.05 92.71 93.40 92.87 92.51 92.80 92.73 92.50 92.82 92.64 92.92 92.96 93.42 92.96 92.66 92.82 93.42 92.74 93.42
256 90.28 86.21 91.31 92.04 91.54 91.66 91.68 92.56 87.29 56.47 91.66 90.80 90.70 92.50 90.89 91.45 92.79 91.68 92.20 89.78 93.88 90.80  -   92.29 90.87
258 90.59 86.96 90.00 90.21 90.26 89.88 90.92 90.93  -   90.78 90.89 90.66 90.69 90.82 90.95 89.51 90.55 91.07 90.78 90.73 90.79 90.29 89.82 90.74 90.86
259 83.00 83.90 86.68 86.69 86.40 86.80 85.72 86.18 86.73 85.27 85.94 86.55 86.37 86.09 86.12 84.78 85.52 86.74 86.63 86.15 86.28 86.27 86.12 86.75 87.02
260 99.91 92.68 99.98 99.97 99.97 85.39 99.91 99.91 99.97 99.73 99.93 99.96 99.96 99.89 99.92 99.62 94.77 99.94 99.96 99.95 99.96 99.95  -   99.96 99.95
263 91.75 91.91 95.23 94.67 93.30 94.61 93.86 93.91 93.99 91.34 93.74 93.93 93.56 93.83 92.96 93.81 93.23 93.88 93.20 93.04 94.35 93.46 94.73 93.40 94.77
264 86.83 94.04 96.17 95.29 95.51 96.31 93.41 94.12 77.30 86.73 68.06 94.93 93.05 94.56 94.90 94.87 51.46 93.34 94.37 92.22 95.78 94.87 85.50 95.60 95.62
265 83.00 83.13 82.40 83.40 82.37 83.77 81.27 82.40 70.17 77.10 83.43 83.30 83.17 82.23 82.17 82.93 83.27 82.53 88.43 83.73 83.53 80.53 84.00 83.30 84.13
266 71.00 75.00 74.03 74.30 72.97 75.13 72.20 70.30 72.27 68.73 69.73 71.23 72.50 71.23 71.57 70.77 68.60 72.40 72.50 70.73 71.57 73.03 73.10 72.40 72.27
268 77.12 72.41 79.71 79.11 77.77 76.90 75.17 76.35 57.48 72.44 74.96 76.32 73.64 67.69 75.65 74.76 28.76 78.63 78.11 74.67 77.53 75.39 76.28 77.99 78.61
272 80.32 80.59 83.99 82.92 83.08 82.68 81.39 82.68 83.21 78.77 81.50 81.04 79.54 83.78 80.72 81.98 80.29 81.90 82.81 83.64 82.33 80.59 82.52 83.78 83.16
273 88.20 81.61 94.54 95.24 93.16 94.24 93.09 93.29 91.54 92.31 93.29 94.71 92.70 92.50 89.94 93.50 91.16 95.88 94.63 95.04 93.30 93.20 94.88
274 97.14 96.77 97.02 97.18 97.17 97.12 96.85 96.67 96.78 96.81 96.86 96.74 96.74 96.62 96.64 96.86 96.60 96.78 96.74 96.62 96.67 96.69 96.92 96.90 96.98
275 77.15 79.59 80.64 80.57 79.65 80.60 79.36 79.56 79.88 78.54 78.90 80.05 79.82 80.02 79.57 78.20 79.02 80.13 79.69 80.31 79.95 80.05 79.70 80.21 80.25
277 95.00 93.82 99.52 99.49 99.64 99.53 98.67 98.50 97.56 98.44 98.42 98.94 98.87 98.40 98.28 93.47 97.13 98.96 99.21 99.10 99.48 99.15 98.56 99.18 99.30
278 95.63 95.71 99.15 99.52 99.29 99.74 98.42 98.08 93.53 97.43 98.14 98.58 98.47 98.07 98.24 98.25 96.12 98.61 99.04 98.99 99.58 98.23 99.43 99.34
279 67.14 69.95 68.93 69.24 69.29 69.69 68.68 68.76 63.50 66.35 68.93 69.32 67.00 69.24 69.60 68.98 68.75 67.39 66.92 68.83 69.06 69.38 69.06 69.06 69.10
282 49.14 46.96 50.66 51.70 51.46 51.47 50.70 50.90 47.51 48.51 50.43 51.53 49.80 50.70 51.28 46.92 47.50 50.85 52.43 51.00 51.63 50.33 49.94 50.96 52.38
283 97.97 97.91 97.63 97.74 97.45 97.57 97.73 97.45 97.91 96.29 97.83 97.82 97.62 96.76 97.81 50.40 64.04 97.62 97.78 97.82 97.85 97.40 97.91 97.77 97.78
285 72.94 75.96 76.04 75.29 72.78 78.12 82.35 84.94 78.20 65.84 80.43 78.24 80.63 84.39 81.10 81.76 75.76 85.76 80.24 81.33 82.31 77.84 84.35 80.90 84.47
288 92.49 68.68 99.58 99.65 99.30 99.82 93.41 93.94 90.96 92.02 92.74 97.97 95.36 92.82 92.72 92.23 90.33 97.73 98.77 98.89 99.30 98.39 94.94 99.22 99.12
289 90.56 89.50 90.25 89.98 90.01 90.41 90.78 91.05 90.06 89.21 90.72 90.13 90.40 90.56 90.25 90.52 90.33 90.50 89.90 90.72 90.92 89.78 89.56 89.60 90.45
290 85.10 86.28 85.41 85.96 85.10 84.95 85.57 86.23 86.25 82.67 86.02 86.11 85.45 85.88 85.32 85.54 40.77 86.04 86.24 86.24 86.70 86.09 85.91 86.29 86.50
291 84.10 86.20 83.69 84.38 82.91 81.73 85.87 86.38 85.69 80.36 85.69 86.42 85.57 85.61 85.67 85.57 85.52 86.26 87.02 85.51 86.27 85.04 86.53 85.59 86.00
293 86.72 86.72 89.86 90.43 89.42 88.17 87.31 88.73 89.62 83.91 88.95 90.14 86.77 90.87 86.81 89.72 87.45 89.50 87.53 88.14 89.64 87.23 89.94 86.96 90.99
295 77.10 70.45 75.00 75.34 73.86 73.39 72.68 72.59 75.17 72.04 71.90 72.34 71.62 72.11 72.05 68.28 74.48 73.93 74.82 72.98 72.90 72.20 73.41 71.81 73.96
297 65.62 54.81 63.62 60.73 63.29 63.60 58.67 58.08 58.23 54.69 50.17 58.19 55.67 56.17 58.52 51.92 61.75 61.27 63.62 60.65 58.75 55.15 58.13 58.29 60.52
298 63.37 53.06 62.42 63.15 62.35 62.95 57.79 57.90 54.30 49.56 54.98 55.17 60.93 55.86 58.84 53.08 31.65 62.65 63.27 59.42 60.81 53.27 57.95 53.49 60.56
299 59.60 58.59 59.84 60.83 62.47 58.25 60.31 60.34 59.57 52.55 59.12 59.82 59.06 59.28 57.53 59.19 16.66 59.84 60.34 59.87 59.26 56.72 59.57 60.88 61.12
```

Table 11: Classification accuracy (mean $\pm$ standard deviation) across 20 high-dimensional datasets. Each method is evaluated over 5 splits and 3 random seeds, and the best-performing method for each dataset is highlighted. "MNCA", "XGB", "RF", "PG" denote "ModernNCA", "XGBoost", "RandomForest", "ProtoGate", respectively.

| Dataset | BETA | MNCA | TabM | MLP | KNN | XGB | RF | PG |
|---|---|---|---|---|---|---|---|---|
| CLL_SUB_111 | $69.86_{\pm6.24}$ | $62.90_{\pm9.10}$ | $72.17_{\pm3.82}$ | $72.46_{\pm6.28}$ | $57.39_{\pm6.96}$ | $\mathbf{73.04}_{\pm7.65}$ | $70.14_{\pm6.32}$ | $65.51_{\pm7.69}$ |
| BASEHOCK | $96.56_{\pm0.77}$ | $96.31_{\pm0.83}$ | $96.89_{\pm0.81}$ | $\mathbf{97.01}_{\pm0.51}$ | $71.88_{\pm4.39}$ | $95.29_{\pm0.79}$ | $96.73_{\pm1.14}$ | $96.32_{\pm2.70}$ |
| Prostate_GE | $88.89_{\pm6.43}$ | $81.27_{\pm8.60}$ | $\mathbf{90.16}_{\pm5.35}$ | $86.98_{\pm6.61}$ | $80.00_{\pm3.56}$ | $89.52_{\pm6.07}$ | $87.94_{\pm7.75}$ | $84.13_{\pm6.19}$ |
| PCMAC | $86.77_{\pm1.26}$ | $88.21_{\pm1.44}$ | $88.02_{\pm1.13}$ | $88.53_{\pm1.26}$ | $66.48_{\pm5.23}$ | $91.64_{\pm1.45}$ | $\mathbf{92.20}_{\pm1.09}$ | $88.21_{\pm3.79}$ |
| GLI_85 | $\mathbf{86.67}_{\pm9.48}$ | $81.57_{\pm8.00}$ | $79.22_{\pm6.02}$ | $85.49_{\pm7.08}$ | $76.47_{\pm10.52}$ | $82.35_{\pm6.08}$ | $83.92_{\pm7.58}$ | $81.96_{\pm9.48}$ |
| RELATHE | $90.19_{\pm1.45}$ | $88.18_{\pm1.42}$ | $89.91_{\pm1.29}$ | $\mathbf{90.54}_{\pm1.70}$ | $75.03_{\pm4.75}$ | $87.11_{\pm1.19}$ | $87.30_{\pm1.28}$ | $89.92_{\pm2.50}$ |
| SMK_CAN_187 | $\mathbf{72.28}_{\pm7.05}$ | $63.51_{\pm6.36}$ | $67.02_{\pm6.99}$ | $66.84_{\pm8.64}$ | $69.47_{\pm5.42}$ | $66.49_{\pm6.62}$ | $70.70_{\pm4.17}$ | $70.71_{\pm5.13}$ |
| warpPIE10P | $\mathbf{100.00}_{\pm0.00}$ | $98.41_{\pm1.66}$ | $\mathbf{100.00}_{\pm0.00}$ | $99.05_{\pm1.17}$ | $92.38_{\pm3.81}$ | $94.92_{\pm3.46}$ | $98.57_{\pm1.17}$ | $97.79_{\pm6.68}$ |
| leukemia | $95.11_{\pm3.82}$ | $90.22_{\pm5.90}$ | $91.11_{\pm4.66}$ | $95.11_{\pm2.95}$ | $86.67_{\pm5.96}$ | $\mathbf{97.78}_{\pm3.14}$ | $92.00_{\pm2.67}$ | $94.00_{\pm10.55}$ |
| orlraws10P | $96.67_{\pm3.50}$ | $97.67_{\pm2.49}$ | $\mathbf{99.00}_{\pm2.00}$ | $98.33_{\pm2.36}$ | $92.0_{\pm2.45}$ | $84.33_{\pm6.80}$ | $\mathbf{99.00}_{\pm2.00}$ | $92.67_{\pm5.73}$ |
| GLIOMA | $\mathbf{72.67}_{\pm9.29}$ | $58.00_{\pm11.08}$ | $62.00_{\pm11.08}$ | $60.67_{\pm12.36}$ | $68.00_{\pm9.80}$ | $66.67_{\pm8.69}$ | $64.00_{\pm14.51}$ | $69.91_{\pm15.78}$ |
| lung_discrete | $72.00_{\pm10.10}$ | $68.00_{\pm8.84}$ | $76.00_{\pm10.83}$ | $71.56_{\pm10.46}$ | $73.33_{\pm5.96}$ | $\mathbf{77.33}_{\pm9.04}$ | $71.56_{\pm8.60}$ | $70.01_{\pm7.38}$ |
| warpAR10P | $91.54_{\pm7.05}$ | $83.08_{\pm6.71}$ | $\mathbf{96.15}_{\pm4.21}$ | $85.64_{\pm6.95}$ | $53.08_{\pm6.15}$ | $81.28_{\pm9.09}$ | $87.18_{\pm5.73}$ | $90.04_{\pm13.95}$ |
| TOX_171 | $\mathbf{91.24}_{\pm3.53}$ | $76.00_{\pm6.33}$ | $87.81_{\pm3.04}$ | $88.19_{\pm4.53}$ | $70.86_{\pm5.54}$ | $78.10_{\pm6.40}$ | $78.67_{\pm5.10}$ | $85.52_{\pm7.41}$ |
| lung | $\mathbf{95.77}_{\pm1.08}$ | $91.54_{\pm3.43}$ | $93.01_{\pm2.8}$ | $95.45_{\pm1.22}$ | $93.66_{\pm1.95}$ | $93.66_{\pm2.64}$ | $92.68_{\pm2.52}$ | $95.43_{\pm4.13}$ |
| ALLAML | $\mathbf{96.89}_{\pm4.12}$ | $87.56_{\pm6.38}$ | $92.00_{\pm2.67}$ | $95.56_{\pm3.14}$ | $81.33_{\pm7.77}$ | $96.00_{\pm4.07}$ | $96.44_{\pm3.33}$ | $91.14_{\pm10.00}$ |
| colon | $\mathbf{85.13}_{\pm5.94}$ | $78.46_{\pm11.65}$ | $82.56_{\pm7.14}$ | $78.97_{\pm9.09}$ | $76.92_{\pm12.87}$ | $74.87_{\pm8.17}$ | $82.56_{\pm6.57}$ | $78.46_{\pm8.04}$ |
| gisette | $97.32_{\pm0.21}$ | - | $97.15_{\pm0.23}$ | $\mathbf{97.57}_{\pm0.32}$ | $95.04_{\pm0.38}$ | $97.55_{\pm0.54}$ | $96.82_{\pm0.61}$ | $97.18_{\pm0.72}$ |
| madelon | $58.29_{\pm2.11}$ | $64.97_{\pm5.79}$ | $56.97_{\pm2.44}$ | $56.44_{\pm1.82}$ | $53.58_{\pm2.74}$ | $\mathbf{75.77}_{\pm2.53}$ | $70.42_{\pm2.30}$ | $72.13_{\pm5.54}$ |
| arcene | $86.67_{\pm4.05}$ | $81.67_{\pm3.84}$ | $78.83_{\pm6.38}$ | $85.50_{\pm3.89}$ | $84.50_{\pm3.32}$ | $75.00_{\pm8.37}$ | $\mathbf{86.83}_{\pm5.44}$ | $85.33_{\pm4.82}$ |

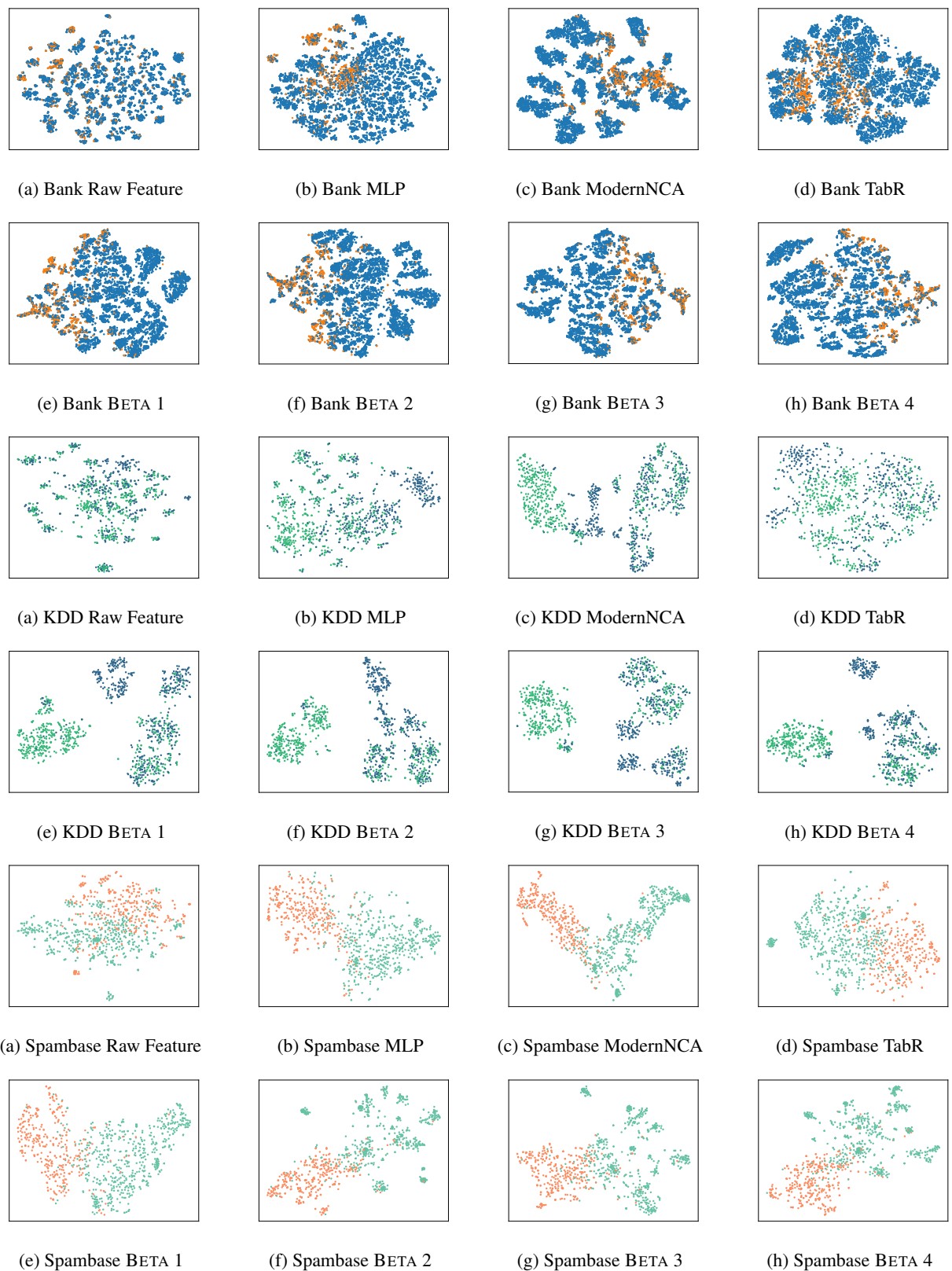

Figure 9: Visualization of the embedding space of different methods.

