# OpenReview forum: "TabPFN Unleashed: A Scalable and Effective Solution to Tabular Classification Problems"
_ICML.cc/2025/Conference — ICML 2025 poster_

### Official Review · Reviewer_2vtp · 2025-02-23

**Overall Recommendation:** 4

**Summary:**

This paper introduces BETA, a method to improve the scalability and performance of TabPFN (a transformer-based technique for tabular classification). BETA combines (1) a lightweight encoder, fine‐tuned (with the pre-trained TabPFN frozen) to re-map raw features into a latent space and better align with downstream data distributions (to reduce bias), (2) multiple encoder paths using a Batch Ensemble strategy to introduce diversity and reduce variance, (3) bootstrapped sampling during inference to generate diverse support and prediction sets, which are aggregated (uniformly or with weights), (4) Error-Correcting Output Codes to extend the model to multi-class classification tasks (beyond 10 classes).

Extensive experiments on benchmark datasets (e.g., TALENT, high-dimensional datasets) and ablation studies, as well as bias-variance analyses, support the claim that BETA improves accuracy and efficiency relative to prior TabPFN variants and other state-of-the-art tabular models.

**Claims And Evidence:**

The claims are supported by comprehensive experiments over 200+ datasets, detailed ablation studies, and analyses of bias-variance decomposition. The experimental results, significance tests (e.g., Wilcoxon-Holm tests), and efficiency metrics (inference time, parameter count) provide empirical support.

While the empirical evidence is good, some claims regarding theoretical guarantees for bias and variance reduction rely on standard decompositions (i.e., the bias-variance decomposition, Eq. (13), is well established in the literature). Additional theoretical justification or analysis could further strengthen the bias and variance reduction claims.

**Essential References Not Discussed:**

There are several recent papers exploring how the rich contextual and semantic representations generated by large language models can be integrated into prediction models for tabular data. Mentioning this stream of work in the introduction would inform and help readers understand the broader context.

**Experimental Designs Or Analyses:**

The experimental setup is robust. The authors use multiple random splits and seeds and compare against a range of competitive baselines (classical and deep learning models). The bias-variance analysis is clearly presented; efficiency metrics (inference time, checkpoint size) are well reported.

Some important experimental details (e.g., hyperparameter ranges and exact training protocols) are in the supplementary material. Including a brief summary in the main text would improve the clarity.

**Methods And Evaluation Criteria:**

The introduced methods are sound. The use of diverse benchmark datasets (e.g., the TALENT benchmark and high-dimensional datasets) along with multi-seed experiments and statistical significance tests is appropriate for evaluating the proposed methods.
A more in-depth sensitivity analysis regarding hyperparameter settings (e.g., number of bootstrapped samples or encoder paths) would enhance the evaluation.

**Other Comments Or Suggestions:**

1. A brief discussion on the computational trade-offs of adding more encoder paths versus performance gains.
2. See weaknesses above.

## After Rebuttal
The authors do address significant and well-known shortcomings of TabPFN and demonstrate consistent improvements on a wide benchmark. However, I do strongly agree with Reviewer 3Q5h that the code should be public. I strongly assume the link to the open-source code will be made available upon publication. Otherwise, there would be a serious replicability limitation. In good faith that the authors will open-source their code and adjust the paper according to the reviews, I would maintain my current score.

**Other Strengths And Weaknesses:**

Strengths:
1. The approach builds on well-established techniques.
2. The experimental evaluation is extensive and covers a wide range of datasets and settings.
3. The integration of ECOC to overcome multi-class limitations is well-motivated.

Weaknesses:
1. The theoretical foundation of the method could be stronger.
2. Some critical implementation details and hyperparameter sensitivity analyses should be summarized in the main part.
3. The limitations section is brief, and further discussion on potential challenges (e.g., distribution shifts, application to regression) would be welcome.
4. More recent work on improving predictions on tabular data through contextual embeddings could be included and discussed.

**Questions For Authors:**

1. Can you elaborate on how the combination of multiple encoder paths specifically contributes to variance reduction? Are there any theoretical bounds or insights that could be provided?
2. How sensitive is the performance of BETA to the choice of the number of bootstrapped samples and encoder paths? Would you consider adding a sensitivity analysis section?
3. You briefly mention the limitation regarding regression tasks. What modifications would be needed to extend BETA to regression problems, and do you have preliminary results in that direction?
4. Have you tested BETA under scenarios with distribution shifts (covariate or concept drift)? If not, could you discuss potential approaches to handle such scenarios?

**Relation To Broader Scientific Literature:**

The paper builds directly on prior work in tabular deep learning, notably TabPFN, TuneTables, and MixturePFN, while incorporating ensemble methods and parameter-efficient fine-tuning techniques known from the broader deep learning literature. There is a strong focus on TabPFN techniques; the combination of ensemble methods with advanced embedding techniques (e.g., https://arxiv.org/abs/2411.01645) could be further analyzed and discussed.

**Theoretical Claims:**

The theoretical aspects used in this paper are well-established in the literature.

---

> ### Author Rebuttal · Authors · 2025-04-01
>
> Thanks for the valuable feedback. In this rebuttal, we address the reviewers' suggestions and concerns in a Q&A format:
>
> ---
> **Q1: Sensitivity Analysis**
>
> ---
> **A1**: We sincerely appreciate the reviewer’s valuable suggestion. To address the sensitivity of BETA’s performance to the **number of bootstrapped samples** and **encoder paths**, we have conducted additional experiments.
>
> - In **Appendix D.3**, we provide an analysis of the **number of encoders** and other key components in BETA.
>
> - Additionally, we include performance rankings for BETA across various datasets from the **Tiny Benchmark2** classification dataset [1], where we vary the **number of bootstrapped samples**. A table summarizing the results is presented below:
>   | size         | 500  | 1000 | 2000 | 3000     | tabpfn |
>   | ------------ | ---- | ---- | ---- | -------- | ------ |
>   | average rank | 3.39 | 2.87 | 2.57 | **1.91** | 4.24   |
> - We also summarize the performance of Beta with different **number of encoders** and their corresponding **average training time** across the datasets of **Tiny Benchmark2**. Specifically, we show the **average percentage change in performance relative to the original TabPFN**.
>   | number of encoders   | 1    | 2    | 4     | 8     | 16    |
>   | -------------------- | ---- | ---- | ----- | ----- | ----- |
>   | relative improvement | 1.7% | 2.6% | 3.19% | 3.23% | 3.76% |
>   | finetuning time (s)  | 31   | 48   | 73    | 126   | 203   |
>
>
> The result demonstrates that both **the number of bootstrapped samples** and **the number of encoders** lead to **better performance**, but they also **increase computational costs**.
>
> We hope these additional details address the reviewer’s concerns. Additionally, we will provide a brief summary of the **experimental details and sensitivity analysis** in the main text of the final version of the paper.
>
> ---
> **Q2: Recent Work on Contextual Embeddings**
>
> ---
> **A2**: We sincerely thank the reviewer for the suggestion. We have noted that the paper referenced by the reviewer, [2], provides a detailed analysis of the **combination of ensemble methods with advanced embedding techniques**. We will incorporate a discussion of this work and related research in the final version of the paper. Additionally, we plan to explore how these techniques could be integrated with BETA in our future work.
>
> We appreciate the reviewer’s insight and will ensure this topic is adequately addressed in the final version of the manuscript.
>
> ---
> **Q3: Extending BETA to Regression Problems**
>
> ---
> **A3**: Thank you for your insightful comment. We applied BETA to regression datasets by using a pre-trained regression version of PFN, where the final classification head has an output dimension of 1, while keeping the other modules unchanged. The only modification was replacing the loss function with MSE loss and fine-tuning it on downstream regression datasets.
>
> We evaluated the regression version of BETA on 100 regression datasets from **[1]** and compared the **average rank** to other tuned methods (we only display a subset of the methods for clarity) presented in **[1]**. Our results show that the regression version performs well, although its performance is slightly lower compared to its classification counterpart.
>
> |Method|catboost|tabr|Beta|FT-T|XGBoost|MLP|KNN|
> |-|-|-|-|-|-|-|-|
> |avg rank|7.3|9.2|9.3|9.8|10.0|12.7|17.8|
>
> We will clarify these findings and potential directions for future work in the final version of the paper. Thank you again for your valuable suggestion!
>
> ---
> **Q4: Response to Testing BETA under Distribution Shifts**
>
> ---
> **A4**: Thank you for your thoughtful question. We have tested BETA on three classification datasets from **TabRed** [3] and integrated the **split method** (Beta-split) and **temporal embedding techniques** (Beta-temporal) discussed in [4] into BETA. The table below presents the **AUC results** of **BETA** on these datasets, showing that the plugin techniques are effective for improving BETA’s performance. Additionally, these results highlight the potential of **ICL methods** for addressing **temporal shift** scenarios.
>
> |dataset|Beta|Beta-split|Beta-temporal|MLP|
> |-|-| -|-|-|
> | ecom-offers|0.5692|0.6264| **0.6348**|0.5866|
> | homesite-insurance|0.9600| **0.9602** |0.9521| 0.9404 |
> | homecredit-default|0.6114| **0.6710** |0.6637| 0.4730 |
>
>
> We will include these findings and further clarify this in the final version of the paper. Thank you again for the valuable suggestion!
>
> [1] A Closer Look at Deep Learning Methods on Tabular Datasets. 2024
>
> [2] Enriching Tabular Data with Contextual LLM Embeddings: A Comprehensive Ablation Study for Ensemble Classifiers. 2024
>
> [3] Tabred: A benchmark of tabular machine learning in-the-wild. 2025
>
> [4] Understanding the Limits of Deep Tabular Methods with Temporal Shift. 2025

---

> > ### Comment · Reviewer_2vtp · 2025-04-01
> >
> > I appreciate the authors’ careful consideration of my concerns and questions. I will increase my score to “accept”.

---

> > > ### Author Response · Authors · 2025-04-08
> > >
> > > Thank you very much for your recognition and support of our work. We are grateful that you found our paper *well-motivated*, with *sound* methods and claims that are *supported by comprehensive experiments*. We sincerely appreciate your thoughtful suggestions, which are very valuable for improving the quality and completeness of our work. Thank you again for taking the time to review our paper.

---

### Official Review · Reviewer_RCWy · 2025-03-14

**Overall Recommendation:** 4

**Summary:**

The authors introduce BETA, a TabPFN variant featuring multiple improvements to the original TabPFNv1 model. BETA introduces encoder-based fine-tuning, multiple encoder fine-tuning, batch-ensemble encoding for inference optimization, inference-time bootstrapped sampling, and an error correcting output code strategy to extend model support beyond 10 target classes. Across over 200 datasets and 15 seeds, BETA outperforms a suite of 28 baseline methods with 0.05 CD significance. BETA also performs well on high-dimensional datasets and maintains TabPFNv1-level inference throughput and a small model artifact of under 1 MB. The authors additional show ablation studies covering variants of TabPFN in Figure 2 with discussion of bias-variance tradeoff impact.

**Claims And Evidence:**

The author's put substantial effort in providing convincing evidence for the claims they make in the paper.

**Essential References Not Discussed:**

The authors do a good job of highlighting relevant works.

**Experimental Designs Or Analyses:**

The experimental design appears solid and sufficiently large-scale to warrant statistically significant take-aways. I do however have some major reservations based on the specifics of how the experiments were conducted to ensure a fair comparison for all methods, which I detail below and look forward to the authors providing clarifications on:

### "Each dataset is randomly split into training, validation, and test sets with proportions of 64%, 16%, and 20%, respectively." & "To ensure a fair comparison, all TabPFN variants, including BETA, are evaluated using their default hyper-parameters without additional tuning"
- What is the validation set used for? TabPFN generally doesn't need to early stop or hyperparameter tune, so are we just throwing away useful data as unused validation data instead of incorporating it into the train_data to fit a better model?
- Is the validation dataset combined with the training data for the final model fit after HPO?

### We report accuracy as the evaluation metric, where higher accuracy indicates better performance.
- Why accuracy? Generally ROC AUC or logloss are more informative. Accuracy can be highly sensitive to the decision threshold, which is not something every model includes as a tunable parameter. Accuracy also doesn't do well in measuring how well calibrated the model is.
- Were the models early stopped to maximize accuracy? For example, LightGBM, XGBoost, RealMLP, etc. can all be early stopped using validation data to optimize a target metric. This would greatly improve their results. Were the fine-tuned TabPFN methods (including BETA) early stopped to optimize accuracy? If so, were these methods ever refit on all of the data afterwards using the found optimal epoch/iteration?

### SOTA Claim & Bagging
- BETA essentially implements internal bagging. Bagging is well known to be virtually universally helpful in improving the strength of models. This also applies to models such as RealMLP, LightGBM, CatBoost, etc. Have the authors considered also fitting bagged versions of the baselines to more faithfully compare top-end performance potential of the methods? I would suspect BETA to at best only marginally improve with explicit bagging (since it already incorporates internal bagging), whereas methods such as RealMLP, CatBoost, etc. will likely benefit greatly from it. This may somewhat change the take-aways in the paper. Note that even if BETA is no longer #1, that does not become a weakness of the paper.

### SOTA Claim & HPO
- BETA and the TabPFN methods are not hyperparameter tuned, however the other methods are. Tuning via a single train/val split is suboptimal and can lead to major overfitting. Tuning via cross-validation is much more preferable and would lead to better results, especially on smaller datasets. Have the authors considered doing this to compare with stronger baselines?

### General comment on experimental design
I suggest the authors read a recent paper detailing shortcomings of many recent tabular benchmark studies, including those that this paper builds its experimental design off of: https://arxiv.org/pdf/2503.09159 . The paper shares similar concerns that I have mentioned above, such as the usage of a holdout set instead of cross-validation.

**Methods And Evaluation Criteria:**

Yes. The evaluation criteria, proposed methods, and comparison baselines all make sense for the target application.

**Other Comments Or Suggestions:**

I agree with the authors that TabPFNv2 is concurrent work, however I think it would elevate the paper if it is possible to include a version of Figure 4 with TabPFNv2 results in the appendix.

Methods that are missing from Figure 4 that could be useful to include:
- TabPFNv2
- TabForest
- TabForestPFN
- TabPFNMix
- Attic
- TabICL
- An AutoML system, as done in the TabPFNv2 paper
- Bagged baseline models

**Other Strengths And Weaknesses:**

I like the usage of Batch Ensemble for compute efficiency. In general the authors apply many clever tricks which are very sensible.

It would have been nice to have an ablation study of iteratively adding the BETA improvements to TabPFN, showing the impact of each component that make up BETA. This would be a very strong addition and would help justify each component (whereas currently it is unknown if certain components are largely superficial / non-impactful).

**Questions For Authors:**

Refer to the other sections. My score is contingent on the notable questions and concerns being addressed. The paper is generally of very high quality, but with some major potential pitfalls that might lead to incorrect conclusions.

**Relation To Broader Scientific Literature:**

The contributions are highly related to the TabPFN and tabular foundation model literature, many of which are recent concurrent works (TabPFNv2, Attic, TabICL, etc.).

**Theoretical Claims:**

While I have not dived deep into the specifics of the theoretical claims, the approaches described make sense and appear reasonable.

---

> ### Author Rebuttal · Authors · 2025-04-01
>
> Thanks for the valuable feedback. In this rebuttal, we address the reviewers' suggestions and concerns in a Q&A format:
>
> ---
> **Q1: What is the validation set used for?  Were the models early stopped to maximize accuracy?**
>
> ---
> **A1**: We follow [1, 2, 5, 6], where the validation set is used only for hyperparameter tuning and early stopping, not for training. For example, for baselines like LightGBM, XGBoost, and FT-Transformer, we use the validation set for early stopping. For BETA and TabPFN variants, we use it for early stopping but do not refit on the combined training and validation set.
>
> ---
> **Q2: Why accuracy?**
>
> ---
> **A2**: We followed the evaluation metrics used in [1, 2, 5, 6]. We also computed AUC-based average ranks (see below). RealMLP's lower AUC ranking may be linked to label smoothing, aligning with findings in [3].
> |Method|avg rank|
> |-|-|
> |Beta|5.1|
> |catboost|7.3|
> |ModernNCA|7.64|
> |LightGBM|8.92|
> |RealMLP|10.04|
> |TabPFN|10.38|
> |FT-T|10.92|
> |MLP|12.19|
>
> ---
> **Q3: Bagging**
>
> ---
> **A3**: We agree that other baselines may benefit from bagging. However, since TabPFN can **change context without retraining**, we only applied inference-stage bagging for BETA. As a result, for other baselines, bagging would **significantly increase the cost**—scaling nearly linearly with the number of base models.
>
> To evaluate bagging’s impact on other models, we trained 16 classifiers per dataset with different random seeds on subsets of size max(0.8 × train_size, 1000) for CatBoost, LightGBM, and RealMLP, ensuring that each subset was at least as large as the ones used by BETA during inference.
> |Method|avg rank|
> |-|-|
> |Beta|6.1|
> |realmlp-bagging|6.97|
> |catboost-bagging|7.42|
> |catboost|8.06|
> |realmlp|8.91|
> |lightgbm-bagging|9.37|
> |lightgbm|10.01|
>
> We appreciate your suggestion and will further enhance the discussion on Bagging in the final version.
>
> ---
> **Q4: HPO**
>
> ---
> **A4**: We followed the hyperparameter tuning approach of [1, 2, 5, 6]. We carefully read the article you referenced [4] and implemented the 5-fold CV evaluation protocol from [4] (Section 2). However, we found this protocol to be significantly **more computationally intensive**.
>
> Due to computational constraints, this approach was applied only to XGBoost, RealMLP, and Beta on classification datasets with fewer than 3,000 rows (from [1]).
>
> For XGBoost and RealMLP, we performed HPO using 5-fold CV, then averaged the predictions from the 5 trained models.
> For BETA, we maintained its default hyperparameters but trained five models on different training folds and averaged their predictions.
>
> The results, shown in the table below, indicate that cross-validation does improve model performance, even for methods like BETA that do not require HPO, though the gain for BETA is relatively modest.
>
> We sincerely appreciate your valuable feedback on HPO strategies, and in the final version of the paper, we will conduct a more detailed evaluation of these effects.
>
> |Method|Avg AUC|
> |-|-|
> |XGBoost|0.8452|
> |XGBoost-cv|*0.8470*|
> |RealMLP|0.8610|
> |RealMLP-cv|*0.8739*|
> |Beta|0.8759|
> |Beta-cv|*0.8785*|
>
> ---
> **Q5: I agree with the authors that TabPFNv2 is concurrent work, however I think it would elevate the paper if it is possible to include a version of Figure 4 with TabPFNv2 results in the appendix.**
>
> ---
> **A5**: In the final version of the paper, we plan to include additional baselines. Due to time constraints, we evaluated TabPFN-v2 and TabICL, both using default settings.
>
> To ensure a fairer comparison, we also tested subsampled variants (TabPFN-sub, TabICL-sub) with 16 training subsets and different preprocessing strategies.
>
> Results show that full versions of TabPFN-v2 and TabICL approach BETA’s performance, while **subsampled versions perform significantly worse**. Our method is currently limited by memory constraints, but a well-optimized TabPFN-v1 could allow larger context sizes, potentially further improving results.
> This suggests that our method is currently constrained by engineering-related memory limitations. If a highly optimized version of TabPFN-v1 were available, we believe our approach could scale to even larger contexts, potentially further improving performance.
>
> |Method|avg rank|
> |-|-|
> |Beta|8.65|
> |PFN-v2|8.76|
> |TabICL|9.16|
> |PFN-v2-sub|12.56|
> |TabICL-sub|12.76|
>
> ---
> **Q6: Sensitivity Analysis**
>
> ---
> **A6**: Please refer to A1 for reviewer 2vtp.
>
> [1] Revisiting deep learning models for tabular data. 2021
>
> [2] A Closer Look at Deep Learning Methods on Tabular Datasets. 2024
>
> [3] Better by default: Strong pre-tuned mlps and boosted trees on tabular data. 2024
>
> [4] Unreflected Use of Tabular Data Repositories Can Undermine Research Quality. 2025
>
> [5] On Embeddings for Numerical Features in Tabular Deep Learning. 2022
>
> [6] TabR: Tabular Deep Learning Meets Nearest Neighbors in 2023. 2023

---

> > ### Comment · Reviewer_RCWy · 2025-04-03
> >
> > Exceptional rebuttal. The authors have addressed nearly all of my concerns and have gone to great lengths to improve their paper by incorporating bagging, cross-validation HPO, TabPFN-v2 and TabICL into the paper.
> >
> > Due to the major improvements, I have increased my score to a 4, and I eagerly await the Beta model release to try it out myself!

---

> > > ### Author Response · Authors · 2025-04-08
> > >
> > > We sincerely thank you for your recognition, encouragement, and kind words. We are especially grateful that you decided to increase your score for our paper. We are truly grateful that you highlighted the strengths of our work, noting that “the approaches described make sense”, “the experimental design appears solid and sufficiently large-scale to warrant statistically significant take-aways”, “many clever tricks which are very sensible”, and that “the paper is generally of very high quality.”
> > > Your thoughtful feedback and suggestions are extremely valuable for further improving the quality of our work. From your comments and the depth of your insights, it is clear that you are an expert in deep learning for tabular data. After acceptance, we will take time to organize our code for better readability and usability, and release it to support the development of tabular foundation models and the broader community.
> > > Thank you again for your time, efforts, and generous support.

---

### Official Review · Reviewer_3Q5h · 2025-03-14

**Overall Recommendation:** 1

**Summary:**

This paper narrows its study to an adaption method for TabPFN, which incorporates a fine tuning encoder, boostrapped sampling in the inference stage into the whole pipeline. The BETA is able to solve the bias and variance and achieve comparable performance.

**Claims And Evidence:**

Yes

**Essential References Not Discussed:**

Some foundation tabular model is missing, such as [1] and [2]

[1] Kim, M.J., Grinsztajn, L. &amp; Varoquaux, G.. (2024). CARTE: Pretraining and Transfer for Tabular Learning. Proceedings of the 41st International Conference on Machine Learning
[2]Yang, Yazheng et al. “UniTabE: A Universal Pretraining Protocol for Tabular Foundation Model in Data Science.” International Conference on Learning Representations (2023).

**Experimental Designs Or Analyses:**

Yes

**Methods And Evaluation Criteria:**

Yes

**Other Comments Or Suggestions:**

1.	The experiments can be carefully designed according to the claims of the proposed method.
2.	More recent studies on foundation model or transfer model should be discussed.

**Other Strengths And Weaknesses:**

Pros:

1.	This work is a new variant of TabPFN, which aims to solve the bias and variance problem in the original TabPFN model. The problem is well motivated by several explorative experiments.
2.	The method incorporates previous techniques to mitigate both bias and variance issues, which is easy to understand.
3.	Experiments demonstrate the effectiveness of the proposed method.

Cons:

1.	The proposed BETA considers encoder to mitigate bias and bootstrapped sampling for variance, which are the techniques already used. Only incorporating both techniques into one pipeline makes this work limited in novelty.
2.	The table 1 gives the comparions between BETA and other variants of TabPFN. However, there is no evidence to show whether the claims are justified.
3.	The experimental design is not so convicing. For example, scale to large datasets, BETA performs worse than LocalPFN on the large scale datasets. But the authors claims that on more other datasets, BETA demonstrates superior performance desipite that this experiment verify the BETA should perform better on large scale datasets. On investigating that BETA is able to reduce bias and variance, the Figure 2 introduce some settings based on TabPFN, not the variants of table 1, which makes the conclusion is not so convicing.

**Questions For Authors:**

Why are the encoders in fine-tuning stage placed before TabPFN instead of after it?

**Relation To Broader Scientific Literature:**

A narrowed contribution to certain model

**Theoretical Claims:**

Yes

---

> ### Author Rebuttal · Authors · 2025-04-01
>
> Thanks for the valuable feedback. In this rebuttal, we address the reviewers' suggestions and concerns in a Q&A format:
>
> ---
> **Q1: Limitation of Novelty**
>
> ---
> **A1**: BETA builds on existing techniques, with its novelty lying in breaking key limitations of TabPFN while maintaining inference efficiency. Simplicity and effectiveness are highly valuable, especially for tabular data:
>
> 1. Overcoming TabPFN’s Limitations:
>    - (a) **Handling Arbitrary Feature Dimensions**: We introduce multiple lightweight encoders, incorporating *batch ensemble* techniques, to map raw features into a unified dimension while keeping TabPFN’s parameters frozen.
>    - (b) **Scaling to Large Dataset**: Bagging extends TabPFN to datasets with more samples.
>    - (c) **Handling Large-Class Problems**: Assigning ECOC codes to different encoders overcomes TabPFN’s class limitations.
> 2. BETA remains compatible with PFN-style batching, ensuring minimal extra model inference computational cost.
> 3. An effective combination of techniques for TabPFN that addresses key weaknesses and significantly outperforms widely used, finely tuned baselines is a meaningful contribution worthy of further study.
>
> ---
> **Q2: But the authors claims that on more other datasets, BETA demonstrates superior performance desipite that this experiment verify the BETA should perform better on large scale datasets.**
>
> ---
> **A2**: This experiment aims to demonstrate BETA’s improved generalization on large datasets. As shown in Figure 5, BETA’s performance gain over TabPFN grows with dataset size.
>
> However, on the largest datasets, LocalPFN performs better because it assigns a unique context to each sample, preventing context sharing and making it **incompatible with the highly parallelizable batching strategy used in PFN**. While this boosts performance, it comes at the cost of much lower efficiency—as shown in Table 3, inference time increases significantly.
>
> LocalPFN is a valuable contribution, but this highlights a trade-off on large datasets: BETA offers better efficiency, while LocalPFN prioritizes local adaptation at a high computational cost.
>
> ---
> **Q3: Figure 2 introduce some settings based on TabPFN, missing the variants of table 1.**
>
> ---
> **A3**: In Section 2.2, we categorize major TabPFN variants into Context Selection and Fine-tuning. As shown in Table 1, all listed variants fall into these two categories.
>
> As stated in Section 2.3, we selected TabPFN-KNN and TabPFN-Finetune as **the most representative methods** for these categories, with other approaches being minor modifications or combinations of these two. To ensure clarity and avoid redundancy, Figure 2 presents results for these key methods.
> The following table shows the relative percentage changes in bias and variance (**negative values indicate reduction**) of Beta and other variants compared to TabPFN, averaged over the dataset sizes used in Figure 2, on the *Adult* and *Bank* datasets. Since TabPFN's absolute variance on these datasets is smaller than its bias, the percentage changes in bias are much smaller than those in variance.
>
> |Adult|Beta|TuneTables|TabForestPFN|LocalPFN|MixturePFN|TabPFN-Bagging|
> |-|-|-|-|-|-|-|
> |Bias|**-4.29** | -1.58 | -2.96 | -3.67 | -3.47 |+1.75|
> |Variance|**-16.52** | +17.93 | +7.76 | +4.38 |+5.26|-10.16|
> |Bank|||||||
> |Bias|**-1.48** |-0.25|-1.12|-1.39|-1.09 |+0.39|
> |Variance |**-39.07**|+25.13|+18.47|+23.02 |+10.35|-9.37|
>
> ---
> **Q4: No evidence to show the claims in Table 1.**
>
> ---
> **A4**: These claims are derived from the specific improvement strategies each method applies to TabPFN. For example, none of the other methods adjust TabPFN’s input representation, meaning they cannot handle high-dimensional datasets. Additionally, evidence supporting the claims about **bias and variance** can be found in *Figure 2* and the table in *A3* above.
>
> We greatly appreciate your suggestion and will further clarify Table 1 in the final version to ensure the claims are well-supported.
>
> ---
> **Q5: Why are the encoders in fine-tuning stage placed before TabPFN instead of after it?**
>
> ---
> **A5**:
>
> 1. Placing encoders before TabPFN allows it to handle **arbitrary feature dimensions** by mapping raw features to a fixed dimension.
>
> 2. Our preliminary explorations found that adding encoders after TabPFN provided almost no improvement.
>
> 3. Moreover, placing them before TabPFN integrates effectively with batch ensemble techniques, enhancing performance.
>
> ---
> **Q6: Some foundation tabular model is missing**
>
> ---
> **A6**: CARTE models tabular data as a graph, embedding column names and entries with a graph-attentional network. UniTabE encodes column names, data types, and cell values as tokens using an encoder-decoder architecture with contrastive learning.
> Unlike TabPFN, which relies on a pre-trained Transformer without semantic modeling, these models incorporate semantic information. We will enhance the discussion on tabular foundation models in the final version.

---

> > ### Comment · Reviewer_3Q5h · 2025-04-04
> >
> > After carefully reading this paper again and all the rebuttals \& responses from other reviewers, the reviewer decides to maintain the initial decision and **strongly recommend rejection**. The reasons go as follows:
> > 1. **No open-source contribution.** This work is good engineering, unleashing and adjusting an existing model, TabPFN. However, it offers no implementation and detailed guidelines for others to reproduce its experiments. Considering the complex structures and many tricks introduced, the reviewer remains highly skeptical about how robust this method is and whether it genuinely outperforms other baselines, as the authors claimed.
> > 2. **Many tricks are introduced into the model without proper justification.** For instance, how does ECOC help with multi-class problems? What are the unique challenges of introducing these designs? What are the benefits? What are the potential drawbacks if this trick is not introduced? Remember Occam's Razor.
> > 3. **The novelty is limited.** It is no secret that ensemble learning can reduce variance. So what's the theoretical contribution in this paper? The reviewer feels there is little. Simplicity and Effectiveness are solid contributions when they are addressing well-motivated problems. But this is not the case for this paper.
> > 4. **There are so many overclaims in the paper.** For instance, *No Additional Inference Cost* in Table 1 is highly misleading. How can ensemble learning have no additional inference cost than naive inference? Additionally, *Handles High-Dimensional Data* is also overclaimed. No evidence supports this claim. In the rebuttal, the author claims *For example, none of the other methods adjust TabPFN’s input representation, meaning they cannot handle high-dimensional datasets.* I don't see the correlation here. If none of the previous literature proves this, the author must conduct empirical experiments to support this strong claim.
> > 5. **There is no takeaway from this paper.** After reading it, the reviewer learned nothing but the hidden message: **This is a good paper, we propose an awesome model, accept us!**. What are the unique pros and cons of adopting TabPFN instead of tree-based or non-pre-trained models? Is this paper addressing these cons? Is the current evaluation solid enough to claim that we are progressing down-to-earth in this domain?
> >
> > To SAC and AC: I am humbly asking you to pay more attention to this paper and make your decision while thinking about the signal sent to the community. If this paper is really accepted by ICML, it will negatively stimulate similar papers into the community. Do we really want researchers to combine tricks and yield on benchmarks instead of reflecting the bigger picture in the community? While both aspects are important, this paper is a bit extreme. I failed to take away any messages or reflection from it. Also, I find it hard to be persuaded by three other reviewers' opinions, even though they **unanimously increased their overall scores to 4.**

---

> > > ### Author Response · Authors · 2025-04-08
> > >
> > > We thank the reviewer for your time for the review process. While we appreciate your feedback, we must respectfully address what appears to be a fundamental misunderstanding of both our work and the TabPFN literature:
> > > > No open-source contribution:
> > >
> > > 1. Thank you for pointing out that “Beta is good engineering” and for acknowledging our efforts to unleash TabPFN.
> > > 2. Regarding your comment that “it offers no ... reproduce its experiments”, we respectfully disagree. **Section 3 provides complete implementation details of Beta, and Appendix B.4 specifies all hyperparameters used in our experiments**. We believe these sections offer sufficient information to support reproducibility.
> > > 3. Regarding robustness, our method outperforms well-tuned baselines on one of the largest tabular benchmarks, as noted by other reviewers. **Detailed ablation studies (Appendix D.3) and additional results (Section A1 in our response to Reviewer 2vtp)** further support its effectiveness and robustness.
> > >
> > > > Many tricks are introduced into the model without proper justification.
> > >
> > > 1. *How does ECOC help...drawbacks without ECOC?*:
> > > Due to architectural constraints, TabPFN and its variants are inherently limited to classification tasks with at most 10 classes (as noted in Lines 355, 974, and 851). To overcome this limitation, ECOC is introduced, enabling Beta to handle >10-class problems.
> > > 2. Regarding Occam's Razor:
> > > Introducing ECOC is necessary to overcome TabPFN's architectural constraint (10-dim logit). It does not violate Occam’s Razor as it is a slight and elegant modification to handle >10-class tasks.
> > >
> > > >  It is no secret ... can reduce variance.
> > >
> > > While bagging may reduces variance, our method is specifically designed to address both bias and variance in TabPFN, which is non-trivial as shown in Figure 2. Most prior variants only improve one side while increasing inference time.
> > >
> > > > Overclaims?
> > >
> > > 1. *No additional inference cost*:
> > > As stated in L128, our comparison is against the **ensemble-style TabPFN commonly used in practice**, not single-pass inference.
> > > This ensemble leverages *PFN-style batching* (Line 82, right), allowing multiple transformed inputs to share context and keeping inference time close to that of the single-pass version.
> > > This stands in contrast to LocalPFN [2], which incurs higher computational costs due to its use of non-shared contexts—leading to an inference count proportional to the size of the test set, incurs additional cost.
> > > Beta follows this same efficient paradigm: for the m encoders in Beta, each encoder performs inference only once, resulting in m predictions. **Thus, our design keeps the high-efficiency benfit of ensemble-style PFN and introduces no additional inference cost compared to ensemble-style TabPFN.**
> > >
> > > 2. *Handles High-Dimensional Data...literature proves this, ...*
> > > Beta’s ability to handle high-dimensional data is an inherent design feature, not an overclaim. TabPFN has a fixed 100D input limit and cannot process >100D data without adaptation (**Line 85, right**).  The reviewer's comment reveals a **lack of understanding of TabPFN's architecture and suggests a lack of careful reading of our paper**.
> > >
> > > > What are the unique pros and cons...instead of tree-based or non-pre-trained models?
> > >
> > > 1. As stated in the Introduction, we have outlined the motivation for studying pre-trained models, which has been thoroughly covered in prior work [1,3,4].
> > > 2. Due to its architectural constraints, TabPFN cannot handle large-scale data (Line 144), high-dimensional features (Line 85), or classification tasks with >10 classes (Lines 355, 974, and 851). Beta addresses these limitations by leveraging bagging, multiple lightweight encoder fine-tuning, and ECOC, each tailored to the specific shortcomings of TabPFN.
> > >
> > > > Do we really want researchers to...bigger picture in the community?
> > >
> > > As the field of tabular foundation models is still in its early stages, contributions like ours help stimulate progress and attract broader research interest by overcoming key architectural limitations of TabPFN, such as its fixed input dimensionality and the 10-class restriction. By addressing these constraints and demonstrating strong empirical performance, our work expands the practical applicability of TabPFN and encourages further exploration and innovation in the development of tabular foundation models.
> > > While our method achieves strong results, it is not merely a collection of tricks aimed at improving benchmarks. Instead, it provides a principled and extensible solution to the design limitations in TabPFN, with implications that go beyond performance metrics and toward broader usability and model design in the tabular foundation model domain.
> > >
> > > [1] Tabpfn: A transformer that solves small tabular classification problems in a second
> > >
> > > [2] Retrieval & fine-tuning for in-context tabular models
> > >
> > > [3] Why In-Context Learning Transformers are Tabular Data Classifiers
> > >
> > > [4] When Do Neural Nets Outperform Boosted Trees on Tabular Data

---

### Official Review · Reviewer_GHpF · 2025-03-20

**Overall Recommendation:** 4

**Summary:**

The manuscript first analyzes the generalization error with a bias-variance decomposition, finding both bias and variance have a non-negligible contribution to the overall error. They then propose an extension tackling both error sources named BETA, which trains multiple dataset-specific encoders (to tackle bias) while averaging the predictions resulting from the different encodings (to minimize variance). Additionally, to further reduce variance during inference, they also use bootstrapped sampling, meaning several random subsets of the training subset as support sets and averaging of the resulting prediction. To scale to more classes, they use Error-Correcting Output Codes.

**Claims And Evidence:**

The claim "Scales to Large Datasets" I am not sure if it is supported, do you mean through the error-correcting output codes? do they affect scaling behavior if there are only two classes? It seems unclear to me how "scales to large datasets" is meant, given "handles high-dimensional data" and "adapts to more than 10 classes" is separately mentioned.

[Update: explained by authors to mean the bagging part]

**Essential References Not Discussed:**

None I am aware of

**Experimental Designs Or Analyses:**

See methods

**Methods And Evaluation Criteria:**

Benchmark itself seems suitable as far as I can judge it.

I am missing an evaluation of the overall computation time, so finetuning time plus inference time compared to other methods, Seems very relevant here to me. [Update: promised by authors to add]

**Other Comments Or Suggestions:**

"To mitigate this, TuneTables (Feuer et al., 2024) offers a more efficient alternative by either fine-tuning the entire model or applying prompt-tuning (Lester et al., 2021), which adjusts only a small set of parameters, reducing resource consumption" -> Formulation seems strange to me, TuneTables only optimizes the prompt, so what does this "either finetuning the entire model" what do you mean here?

[Update]: explained by authors. thanks

"Minimizing L_total in Equation 6 ensures that the model jointly trains all encoders to generate diverse yet TabPFN compatible latent representations, thus reducing variance and providing more stable predictions."
I don't see how it "ensures" "diverse representations", there is nothing enforcing that as far as I see, so ensure seems a misleading formulation here (as one could imagine losses that would ensure it).

In Fig.3 , "Inference Stage" is not centered in the box, also "Mean"/"Loss" isn't? [Update: change promised, thanks]

Figure 5,  I think the x axis is just confusing in this way, it should represent actual dataset sizes, and then maybe one could just make a scatter plot of dataset size vs relative improvement  (notice also typo in y axis).  [Update: change promised, thanks]

**Other Strengths And Weaknesses:**

See below

**Questions For Authors:**

-

**Relation To Broader Scientific Literature:**

Results on one method to further improve tabular in context prediction and marry it with partial finetuning.

**Theoretical Claims:**

None

---

> ### Author Rebuttal · Authors · 2025-04-01
>
> Thanks for the valuable feedback. In this rebuttal, we address the reviewers' suggestions and concerns in a Q&A format:
>
> ---
> **Q1: Finetuning time plus inference time compared to other methods.**
>
> ---
> **A1**: In **Table 3**, we have provided a comparison of **inference time, average rank, and the number of learnable parameters**.
>
> To address this point more comprehensively, we include the **fine-tuning/training** time for **BETA** as well as other methods listed in Table 3. Notably, since **Beta only fine-tunes the encoder parameters**, it has a clear advantage in both **fine-tuning and inference efficiency** compared to other variants.
>
> | Methods                      | Beta  | LocalPFN | MixturePFN | FT-T | TabR |
> | ---------------------------- | ----- | -------- | ---------- | ---- | ---- |
> | finetuning/training time (s) | 62.29 | 186.73   | 83.14      | 58.9 | 121  |
> | inference time (s)           | 0.91  | 12.51    | 3.24       | 0.36 | 0.38 |
>
> Additionally, it is important to highlight that methods like **FT-T and TabR** underwent an extensive hyperparameter tuning process in [2], spanning **100 trials**. As a result, their actual tuning time is approximately **100 times** their reported training duration.
>
> ---
> **Q2: Claim: Response to "Scales to Large Datasets"**
>
> ---
> **A2**: We refer to datasets with a **large number of samples** as **large datasets**.
>
> TabPFN is inherently limited by its **attention mechanism over instances**, leading to high memory overhead. As a result, it typically requires subsampling the training set to remain computationally feasible. To mitigate the impact of context size on model performance during inference, we introduce bagging for TabPFN.
>
> Additionally, since TabPFN is pretrained only on synthetic datasets with fewer than 1000 rows, we incorporate multiple lightweight encoders to reduce bias, enabling the model to better scale to large datasets.
>
> ---
> **Q3: About TuneTables: Formulation seems strange to me, TuneTables only optimizes the prompt, so what does this "either finetuning the entire model" what do you mean here?**
>
> ---
> **A3**: In **classification tasks**, TuneTables employs **prompt tuning**, where *“which adjusts only a small set of parameters, reducing resource consumption”* specifically refers to prompt tuning. However, in their **extension to regression tasks**, they adopt **end-to-end fine-tuning**. This is why we mentioned *"fine-tuning the entire model."* We will clarify this distinction in the final version.
>
> ---
> **Q4: I don't see how it "ensures" "diverse representations", there is nothing enforcing that as far as I see, so ensure seems a misleading formulation here (as one could imagine losses that would ensure it).**
>
> ---
> **A4**: We appreciate the reviewer’s observation. The goal is to ensure that each encoder generates **TabPFN-compatible latent representations**, while diversity is encouraged through different initializations.
>
> Additionally, **TabM** [1] has discussed diversity in base models, and we further analyze BETA’s encoder diversity through **embedding visualizations in Appendix Figure 9**, showing that the learned representations exhibit diversity. We will provide a detailed explanation of **diversity** and **TabPFN-compatible latent representations** in the final version.
>
> ---
> **Q5: Response to Figure Presentation.**
>
> ---
> **A5**: We appreciate the reviewer’s helpful suggestions.
>
> - **Figure 3**: We will adjust the alignment of **"Inference Stage"**, **"Mean"**, and **"Loss"** to ensure proper centering within their respective boxes.
> - **Figure 5**: We will modify the **x-axis** to directly represent **actual dataset sizes** and consider using a **scatter plot of dataset size vs. relative improvement** for better clarity. Additionally, we will correct the **typo in the y-axis**.
>
> These improvements will be incorporated into the final version.
>
> We sincerely appreciate the reviewer’s **careful and constructive feedback**. We believe these valuable suggestions will **further enhance the quality** of our paper, and we will incorporate the necessary revisions accordingly.
>
> [1] Tabm: Advancing tabular deep learning with parameter-efficient ensembling. 2025.
>
> [2] A Closer Look at Deep Learning Methods on Tabular Datasets. 2024.

---

> > ### Comment · Reviewer_GHpF · 2025-04-02
> >
> > Thank you for your explanations and the promises of the manuscript improvements, I have updated my review accordingly.
> > I would still like a change of wording in the "ensure [...] diverse representations" part, e.g. to something like
> > "Minimizing L_total in Equation 6 jointly trains all encoders to generate TabPFN-compatible latent representations, with the different encoder initializations encouraging diverse representations and thereby reducing variance and providing more stable predictions."

---

> > > ### Author Response · Authors · 2025-04-08
> > >
> > > We greatly appreciate the score improvements from the reviewer. We are glad that our response was able to address your concerns, and thank you for recognizing our efforts.
> > > It is clear from your detailed comments and thoughtful responses that you carefully read and thoroughly considered our work. We truly appreciate your responsible and constructive reviewing. Your suggestions have been very helpful in improving the quality of our paper.
> > > We will revise the final version to incorporate your suggestions, especially by improving the clarity of the “ensure [...] diverse representations” part as you pointed out.
> > > Once again, thank you for your time, effort, and valuable feedback.

---

### Decision · Program_Chairs · 2025-05-01

**Decision:**

Accept (poster)

**Comment:**

This submission introduces a generalization of the TabPFN model, a pretrained model for tabular learning. The work analyze that the predictor error comes from both bias and variance, and the TabPFN generalization combines multiple ideas to address them, such as bagging, a better fine-tuning strategy and better multiclass support. The submission generated much interest from the reviewers and discussion. The reviewers appreciated the performance improvements and the evaluation, including the work during the rebuttal to add a comparison to work released after the ICML deadline.

There was a discussion amongst reviewers and AC on the importance of open sourcing the code underlying the present work. The AC would like to stress the need for open sourcing once the work is accepted.